# Evolutionary insights from profiling LINE-1 activity at allelic resolution in a single human genome

Lei Yang [ID] [1,3], Genevieve A Metzger[1,3], Ricky Padilla Del Valle [ID] [1,2], Diego Delgadillo Rubalcaba[1] & Richard N McLaughlin Jr [ID] [1,2 ✉]

## Abstract

**Transposable elements have created the majority of the sequence in many genomes. In mammals, LINE-1 retrotransposons have been expanding for more than 100 million years as distinct, consecutive lineages; however, the drivers of this recurrent lineage emergence and disappearance are unknown. Most human genome assemblies provide a record of this ancient evolution, but fail to resolve ongoing LINE-1 retrotranspositions. Utilizing the human CHM1 long-read-based haploid assembly, we identified and cloned all full-length, intact LINE-1s, and found 29 LINE-1s with measurable in vitro retrotransposition activity. Among individuals, these LINE-1s varied in their presence, their allelic sequences, and their activity. We found that recently retrotransposed LINE-1s tend to be active in vitro and polymorphic in the population relative to more ancient LINE-1s. However, some rare allelic forms of old LINE-1s retain activity, suggesting older lineages can persist longer than expected. Finally, in LINE-1s with in vitro activity and in vivo fitness, we identified mutations that may have increased replication in ancient genomes and may prove promising candidates for mechanistic investigations of the drivers of LINE-1 evolution and which LINE-1 sequences contribute to human disease.**

**Keywords** Transposable Elements; LINE-1; Genome Evolution; CHM1; Structural Variation
**Subject Categories** Chromatin, Transcription & Genomics; Evolution & Ecology; Genetics, Gene Therapy & Genetic Disease

## Introduction

Genomes are plagued by both infectious and endogenous parasites. Among these interlopers, transposable elements have left an indelible mark on most genomes by their recurrent and persistent integration-coupled replication (Smit et al, 2013; Jurka et al, 2005). For example, transposable elements created more than half of the modern human genome and more than 90% of some other vertebrate genomes (Meyer et al, 2021). The Long INterspersed Element-1 (LINE-1 or L1) retrotransposons are the most prolific contributors of new sequences in the human genome. LINE-1s are the only group of autonomous elements with ongoing detectable activity in human genomes (Lander et al, 2001), and as a result of their mutagenic potential, LINE-1s contribute to a slew of human diseases (Hancks and Kazazian, 2016; Payer and Burns, 2019).

To maintain their fitness, LINE-1s must make enough new copies in the germline genome to outpace their rate of acquiring inactivating mutations. Unlike infectious retroviruses, which transmit horizontally and integrate primarily in somatic tissues, LINE-1s (and other endogenous retroelements) transmit vertically (Boissinot and Sookdeo, 2016) (with rare, notable exceptions (Ivancevic et al, 2018; Zhang et al, 2020)) and must, therefore, retrotranspose in the germline. This obligatory germline integration leaves a genomic fossil record of LINE-1 retrotranspositions in the genome of each host and their descendants. As a result, hundreds of thousands of LINE-1 remnants make up ~17% of human genomes and encapsulate the recent and ancient evolutionary history of these elements (Smit et al, 2013; Jurka et al, 2005; Lander et al, 2001; de Koning et al, 2011). For example, LINE-1s that were active in the last mammalian common ancestor over 100 million years ago are still clearly identifiable in extant human genomes (Khan et al, 2006). Phylogenetic analyses of these and other ancient sequences suggest that LINE-1 evolution in the ancestors of humans has been typified by a cycle of emergence, expansion, and death of LINE-1 lineages, with the downfall of an old lineage following the emergence of a new lineage (Khan et al, 2006). One model suggests that competition among contemporaneous LINE-1 subfamilies could underlie these waves (Boissinot and Furano, 2001), but the drivers of these evolutionary transitions remain largely unknown.

In addition to sequences created by ancient (now dead) LINE-1s, the human genome also contains young and currently active LINE-1s (Sassaman et al, 1997; Brouha et al, 2003; Badge et al, 2003). These sequences reflect the most recent bouts of activity and evolution within human LINE-1s. Previous analyses estimated that among the approximately one million identified LINE-1 sequences in the reference human genome, only 3,000-5,000 are full-length (>6 kb) (Lander et al, 2001; Schneider et al, 2017). Subsequent studies found that a large portion of the structural variation among human genomes traces back to the activity of LINE-1s including many polymorphic mobile element insertions (MEIs) within the human population (Beck et al, 2011; Ewing and Kazazian, 2011;

[1]Pacific Northwest Research Institute, Seattle, WA, USA. [2]Molecular and Cellular Biology Graduate Program, University of Washington, Seattle, WA, USA. [3]These authors contributed equally: Lei Yang, Genevieve A Metzger. ✉E-mail: rmclaughlin@pnri.org

Macfarlane et al, 2013; Streva et al, 2015; Chaisson et al, 2019; Ho et al, 2020). Using both whole genome and more targeted sequencing approaches (Beck et al, 2010; Huang et al, 2010; Iskow et al, 2010; Ewing and Kazazian, 2010; Stewart et al, 2011), many studies focused largely on variation in the presence/absence state (also known as "insertion polymorphisms") of MEIs including LINE-1s in individual genomes. Some groups have reported hemizygosity or even sequence differences between the alleles of a specific LINE-1 insertion (Lutz et al, 2003; Seleme et al, 2006; Gardner et al, 2017; Chaisson et al, 2015). However, accurate sequencing of polymorphic LINE-1s within individual genomes has proven difficult, largely due to the fact that reads shorter than the length of LINE-1 cannot be confidently assigned to a specific locus or allele (Chaisson et al, 2015).

Intriguingly, some recent datasets have successfully used high-coverage (Chuang et al, 2021) and long-read (Zhou et al, 2020) sequencing to determine the polymorphic state and sequence of all LINE-1s in the genome of a homozygous cell line derived from a complete hydatidiform mole (CHM) (Chaisson et al, 2019). These represent scalable methods to catalog LINE-1 locations and sequences in individual human genomes.

Even with extensive lists of LINE-1 sequences from numerous genomes, a LINE-1's sequence alone is currently insufficient to predict its in vitro or in vivo retrotransposition activity. Previous studies used an early draft of the human reference genome to identify full-length LINE-1s which were subsequently cloned and tested for their ability to retrotranspose in vitro. Only six of the cloned LINE-1s were highly active or "hot" in vitro, suggesting a small set of young (usually polymorphic) LINE-1s generate the vast majority of the retrotransposition activity in this genome (Brouha et al, 2003). The subsequent identification of numerous population- or individual-specific, highly active LINE-1s (Brouha et al, 2003; Beck et al, 2010) suggested this estimate of six "hot" LINE-1s per genome may not reflect the abundance and activity of LINE-1s in more diverse, representative populations. In addition to sequence and activity variation among different LINE-1s, sequence variation in the same LINE-1 from different genomes can also profoundly affect in vitro retrotransposition activity (Seleme et al, 2006; Sanchez-Luque et al, 2019).

In some cases (Miki et al, 1992; Kimberland et al, 1999; Scott et al, 2016), specific LINE-1 sequences at specific genomic locations have been shown to be causal for certain cases of human disease. The highly polymorphic state of LINE-1s among humans suggests the presence of specific disease-causing LINE-1s also varies among individuals. A fully resolved individual genome sequence could be used to identify the specific locations and sequences of LINE-1s that might predispose an individual to specific diseases (a "risk score," reflecting the activity of LINE-1s found in an individual genome). To achieve such a precise quantification of LINE-1 activity and its associated disease risk would require two major advancements—first, the ability to catalog the precise location and sequence of each LINE-1 in an individual's genome, and second, the ability to predict the activity of LINE-1s based on their sequence and location.

In this paper, we have defined the complete repertoire of intact and retrotransposition-competent LINE-1s in a single human genome. Our approach identified a higher number of active LINE-1s than previous approaches with related genomes. We used existing databases of structural variation to show that many highly active LINE-1s are polymorphic in the human population, but some are fixed, suggesting they have persisted in their active state since the last common ancestor of humans. We further demonstrated that some groups of young and old LINE-1s that are active in in vitro retrotransposition assays have also recently retrotransposed in humans, supporting their activity in vivo, as well. In many cases, specific LINE-1s vary in their sequence and activity among humans, suggesting that each individual may have a divergent set of highly active old LINE-1s in their genome which varies greatly among individuals. Finally, we identified sequence changes that correlate with the high in vivo fitness of certain groups of LINE-1s, some of which may represent determinants of persistent activity or targets of host restriction mechanisms. These findings demonstrate that LINE-1 polymorphisms (hemizygosity and homozygosity with allelic variation) are more complicated than previously thought, reinforcing the notion that the reference genome does not encompass the complete set of active LINE-1s in any individual. This definition of the complete set of active LINE-1s in a haploid genome will allow us to further define the selective pressures and evolutionary trajectories of LINE-1s in human genomes, and lays the foundation for predicting activity from LINE-1 sequences.

# Results

## Comprehensive catalog of intact LINE-1s in homozygous human genomes

The CHM1 (complete hydatidiform mole 1) assembly (Steinberg et al, 2014) represents the nearly homozygous genome (<0.75% heterozygosity) (Fan et al, 2002) of a human hydatidiform mole cell line derived from a European individual. Complete hydatidiform moles form from an ovum, empty of maternal DNA, that has been fertilized by a typical haploid sperm. A single replication yields a genome homozygous for the paternal genotype with two nearly-identical sets of the X chromosome and all 23 autosomal chromosomes. The genome of this human cell line was deeply sequenced with PacBio reads (54×), and the location and type of structural variants, including LINE-1s, found in the resulting assembly have been extensively annotated (Steinberg et al, 2014; Fan et al, 2002; Huddleston et al, 2017). These published analyses mostly focused on the presence of LINE-1 insertions relative to the GRCh38 human reference genome, but the nature of these genomes and assemblies enabled us to confidently retrieve the sequence of each LINE-1, which was not necessarily possible with previous assemblies which often resolved LINE-1 insertion sites but not the sequence of the inserted LINE-1. To collect the complete set of all LINE-1 sequences in CHM1, we extracted all sequences annotated as LINE-1 from the RepeatMasker (Smit et al, 2013) analysis of the assembly. RepeatMasker found 919,967 sequences annotated, comprising ~504 Mbp of total sequence. This corresponds to ~16.82% of the total CHM1 genome, comparable to previous estimates of LINE-1-derived sequence in GRCh38 (~one million LINE-1s and ~540 Mbp) (Lander et al, 2001; Chaisson et al, 2015). Additionally, we examined a newer version of the CHM1 assembly (GCA_001297185.2, v2) to check whether this version contained additional LINE-1s compared to the version of assembly that we initially used (GCA_001297185.1, v1). We BLASTed 2 kb regions

flanking each full-length LINE-1 in the v2 genome against v1 and found the full-length LINE-1 counterpart in the v1 assembly for each case, suggesting that v2 assembly of CHM1 did not add additional full-length LINE-1s.

Active LINE-1 sequences contain two open reading frames (ORFs) which encode proteins (ORF1p and ORF2p) required for replication (Moran et al, 1996). We assume that full-length and intact ORFs are necessary but not sufficient for LINE-1s to be active. To identify putatively-active LINE-1s in the CHM1 genome, we determined the longest continuous ORFs present in each full-length (>5000 bp) LINE-1 that, when translated, aligned to sequences of ORF1p and ORF2p from a reference LINE-1, L1$_{RP}$ (GenBank accession number AF148856) (Kimberland et al, 1999) (Figs. EV1 and EV2). Most LINE-1s longer than 5,000 bp encode truncated ORFs, but many retain ORFs of the same length as the ORFs in L1$_{RP}$ (338 codons for ORF1 and 1,275 codons for ORF2; Fig. EV2C). However, we also included sequences with different ORF lengths that still align along the entire length of the amino acid sequence of the reference, from start to stop codon without terminal deletions or extensions. In this way, we defined "intact" LINE-1s as sequences greater than 5000 bp which contain two intact open reading frames that align, when translated, to the full length of L1$_{RP}$ ORF1p and ORF2p. Using this definition, we identified 148 intact LINE-1s in the assembly of the CHM1 genome.

A second human hydatidiform mole cell line (constructed similarly to CHM1) has been sequenced with 52× PacBio read depth, and the resulting assembly (CHM13) has also been extensively analyzed for structural variants (Huddleston et al, 2017; Schneider et al, 2017). This mole is from an individual of unknown ethnic origin, but clusters with CHM1 and other European genomes. Our computational pipeline found the distribution of LINE-1 sequence lengths in CHM13 was similar to that of CHM1 and identified 142 intact LINE-1s in the CHM13 assembly (Fig. EV3).

While these assemblies provide a greatly-improved resolution of repeat sequences like LINE-1, sequencing and assembly errors could still contribute to an underestimation of the true number of intact LINE-1s. In particular, PacBio sequencing errors most often occur as indels in homopolymer tracts (Hebert et al, 2018), a sequence pattern present at several locations in the LINE-1 sequence. To assess the prevalence of sequencing errors that disrupted true intact LINE-1s, we identified sequences that deviated from an intact LINE-1 sequence by a single frame-shifting mutation (82 LINE-1s) or two frame-shifting mutations (47 LINE-1s). We assumed that frame-shifting mutations shared by any two assemblies (CHM1, CHM13, hg38) were not likely to be the result of sequencing errors but found 27 of these LINE-1s had frame-shifting mutations unique to CHM1 (24 with single frame-shifting mutations and 3 with two frame-shifting mutations). We took advantage of a publicly available BAC library of the CHM1 genome (CHORI-17, The BAC clones from the hydatidiform mole were created at BACPAC Resources by Dr. Mikhail Nefedov and Dr. Pieter J. de Jong using a cell line created by Dr. Urvashi Surti. available at BPRC, https://bacpacresources.org) to Sanger sequence these apparently frame-shifted LINE-1s. While most of the frame-shifting mutation present in the assembly were confirmed, we identified sequencing errors in four CHM1 LINE-1s (with one frame-shifting mutation in the published CHM1 assembly) which revealed them to be intact. None of the LINE-1s containing two annotated frame-shifting mutations in the assembly were intact by

Sanger sequencing. In a complementary analysis, we leveraged a new, high fidelity CHM13 assembly based on circular consensus long reads (Vollger et al, 2020). Using this high-fidelity assembly to check the sequence of 87 LINE-1s in the original CHM13 assembly containing one frame-shifting mutation and 45 LINE-1s containing two frame-shifting mutations, we identified two additional intact LINE-1s (both from the single-shift mutation pool).

In addition to sequencing mutations, we also expected evidence of CHM assembly errors or gaps that excluded intact LINE-1s. Since the donors for CHM1 and CHM13 genomes clustered together with other European genomes, we expected LINE-1s present in only one of these closely-related genomes to have low or intermediate frequencies in the human population. However, we observed two LINE-1s that were present in the CHM13 assembly, absent from the CHM1 assembly, and apparently present in all other human assemblies in the KGP (Auton et al, 2015). Indeed, these LINE-1s fell in regions of the assembly with a high density of contig junctions; Sanger sequencing of these regions from corresponding BACs found that both of these LINE-1s are indeed present and intact in CHM1.

Finally, the flanking sequence of some intact LINE-1s could not be located in the reference genome (GRCh38) using the UCSC genome browser liftover tool. We were also unable to find these unplaced LINE-1s a more complete assembly of this genome—the draft telomere-to-telomere (T2T) CHM13 assembly (Miga et al, 2020; Logsdon et al, 2021; Nurk et al, 2022). Together, our detailed re-analysis of intact LINE-1s in multiple assemblies and targeted resequencing of LINE-1s identified 154 intact LINE-1s in CHM1 (Dataset EV4) and and 144 intact LINE-1s in CHM13 (Dataset EV5).

## A pseudo-diploid genome reveals LINE-1 allelic variation

Clearly LINE-1s are a major contributor to human genome variation; however, we lack a thorough understanding of the scale of LINE-1 insertion and allelic heterozygosity within and between individual genomes. LINE-1 insertions, especially young insertions, are highly polymorphic in the human population (Iskow et al, 2010; Tang et al, 2017; Sudmant et al, 2015; Xing et al, 2009). Some LINE-1 insertions are also hemizygous, meaning a LINE-1 is only present in one of the two chromosomes/haplotypes at a given locus in an individual. In individuals homozygous for a given LINE-1 insertion, variation in the sequence of the two alleles of that LINE-1 has also been reported (Lutz et al, 2003; Seleme et al, 2006) We used the CHM1 and CHM13 assemblies to make genome-wide estimates of the content and variation of LINE-1 that could exist within diploid genomes. We found intact LINE-1s present at 194 distinct loci in the union of these two haploid genomes (Table EV1). Most of these LINE-1s were present in both CHM genomes (131/154 in CHM1 and 121/144 in CHM13), but not necessarily intact in both genomes (Fig. 1A); and 102 were intact in both CHM1 and CHM13 (Fig. 1A, purple shaded area). Amongst the LINE-1s that are present in both genomes, 29 LINE-1s intact in CHM1 are present in CHM13 but have accumulated ORF-disrupting mutations, and 19 LINE-1s are intact in CHM13 but contain ORF-disrupting mutations in CHM1 (Fig. 1A). Finally, despite the relatedness of these two genomes, there are 23 LINE-1s in CHM1 and another 23 in CHM13 that are unique to each genome, respectively (Fig. 1A). All loci with unique LINE-1s are hemizygous and likely represent the youngest LINE-1s in these two genomes.

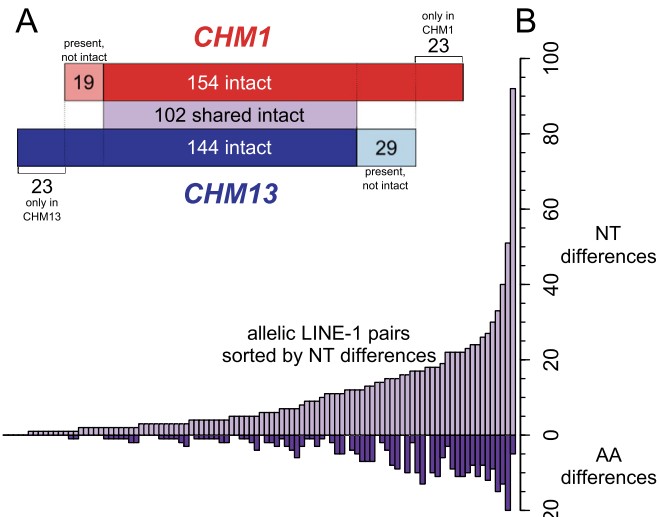

**Figure 1. A comparison of intact allelic LINE-1 pairs across two haploid genome assemblies.**

(A) A comparison of the intact LINE-1s in two nearly homozygous genomes (CHM1, top/red, and CHM13, bottom/blue) based on liftover found 102 shared intact LINE-1s (light purple shaded area). Of the LINE-1s that are intact in only one genome, some are present but not intact in the other genome (light red and light blue shaded areas), and some are absent in the other genome. (B) Distribution of nucleotide (top) and non-synonymous (bottom) changes between CHM1 and CHM13 of the 102 allelic intact LINE-1s ordered by nucleotide differences. NT nucleotide, AA amino acid.

Even though around half of all LINE-1 loci contained an intact LINE-1 in both CHM1 and CHM13, at almost all (98/102) of these loci we observed nucleotide sequence differences between these allelic intact LINE-1s when comparing CHM1 and CHM13. Some diversity of LINE-1 alleles has been described (Lutz et al, 2003), but the CHM assemblies provided an unprecedented opportunity to quantify this form of diversity across an entire genome. At the nucleotide level, some of the largest allelic differences arise from large deletions in the UTRs in one genome; these are scored as multiple single nucleotide differences in the alignment, but often represent single deletion events. Other pairs have deletions and SNPs distributed throughout the UTRs and coding sequence, including one pair with 92 total nucleotide differences. Within ORF1p and ORF2p, 63/102 allelic LINE-1 pairs are identical at the amino acid level, and the most distinct allelic pairs differ by 20 amino acids (Fig. 1B). We found this level of allelic variation surprising and wanted to confirm the sequence difference by Sanger sequencing the CHM1 allele of the four most-divergent LINE-1 pairs. In each case, our resequencing perfectly matched the assembly sequence, suggesting that allelic differences at LINE-1-containing loci are pervasive in human genomes.

## Comprehensive measurement of LINE-1 in vitro activity in a human genome

While these intact LINE-1s contained the sequence characteristics consistent with activity, many likely acquired inactivating synonymous or UTR mutations for which we do not currently have the understanding of LINE-1 sequence:function relationships to predict. Accordingly, a precise measure of the true number of active LINE-1s within a single genome remains a fundamental, unresolved question in the LINE-1 field. Seminal studies based on the draft human genome reference found six highly active LINE-1s (Brouha et al, 2003), though it is now clear that version of the reference genome was biased towards high frequency, older LINE-1s (Beck et al, 2010). Subsequent studies targeting LINE-1s not present in the human reference genome found many active elements among six diverse human genomes and estimated fourteen highly active LINE-1s in one of these individuals. The CHM1 genome's resolution of intact LINE-1s and its associated publicly-available BACs made it possible for us to survey the number of active LINE-1s within a single genome with unprecedented resolution. As a first step, we set out to measure the in vitro retrotransposition rate (hereafter referred to as in vitro activity) of all intact LINE-1s in the CHM1 genome by cloning them from BACs into a retrotransposition reporter construct and testing their activity in an established cell-based assay (Moran et al, 1996; Xie et al, 2011). While some intact LINE-1s were on unplaced contigs (likely within simple or centromeric repeats), we successfully cloned 142/145 intact LINE-1s that could be mapped to specific CHM1 BACs by PCR amplifying the complete LINE-1 sequence (including UTRs) as defined by RepeatMasker and inserting them into a retrotransposition vector with a luciferase reporter. We then transfected three independent clones of each LINE-1 into 293 T cells and measured the transfection-normalized luciferase signal to determine the retrotransposition level of each clone. We compared retrotransposition of each test clone to a commonly studied and highly active human LINE-1, $L1_{RP}$ (Kimberland et al, 1999). As additional controls, we also included a mutant LINE-1 lacking activity (JM111, ORF1p R261A/R262A (Moran et al, 1996)) and an empty vector without a LINE-1.

The majority of the intact LINE-1s from CHM1 had no detectable in vitro retrotransposition activity, consistent with previous activity studies (Brouha et al, 2003; Beck et al, 2010) and our understanding of LINE-1 mutation accumulation. However, we found 29 LINE-1s from CHM1 with measurable retrotransposition activity, defined as >5% $L1_{RP}$ activity (of note, 26 LINE-1s were >10% $L1_{RP}$). Of the 29 active elements, two had significantly higher activity than $L1_{RP}$ and three elements had activity comparable to that of $L1_{RP}$ (Fig. 2). These data suggest that human genomes contain an unappreciatedly large pool of active LINE-1s.

## Comparison of the set of active CHM1 LINE-1s to other genomes

The CHM1 genome contains two of the six previously described "hot" LINE-1s (Dataset EV1, column AB, AC021017 and AC002980) (Brouha et al, 2003); both of these LINE-1s were also highly active in our assays (30.6% and 69.6% of $L1_{RP}$). CHM1 also contains three LINE-1s shown to be polymorphic in a recent study (Beck et al, 2010), and all exhibit high activity (11.6%, 44.5%, 19.7% of $L1_{RP}$, Dataset EV1, column S). Many LINE-1s that were highly active in our analyses were reported as weakly active or dead in another study. For example, the two of the most active LINE-1s in our analysis (261.4% and 86.7% of $L1_{RP}$, Dataset EV1, column S) were both previously studied but found to be very weakly active (Brouha et al, 2003).

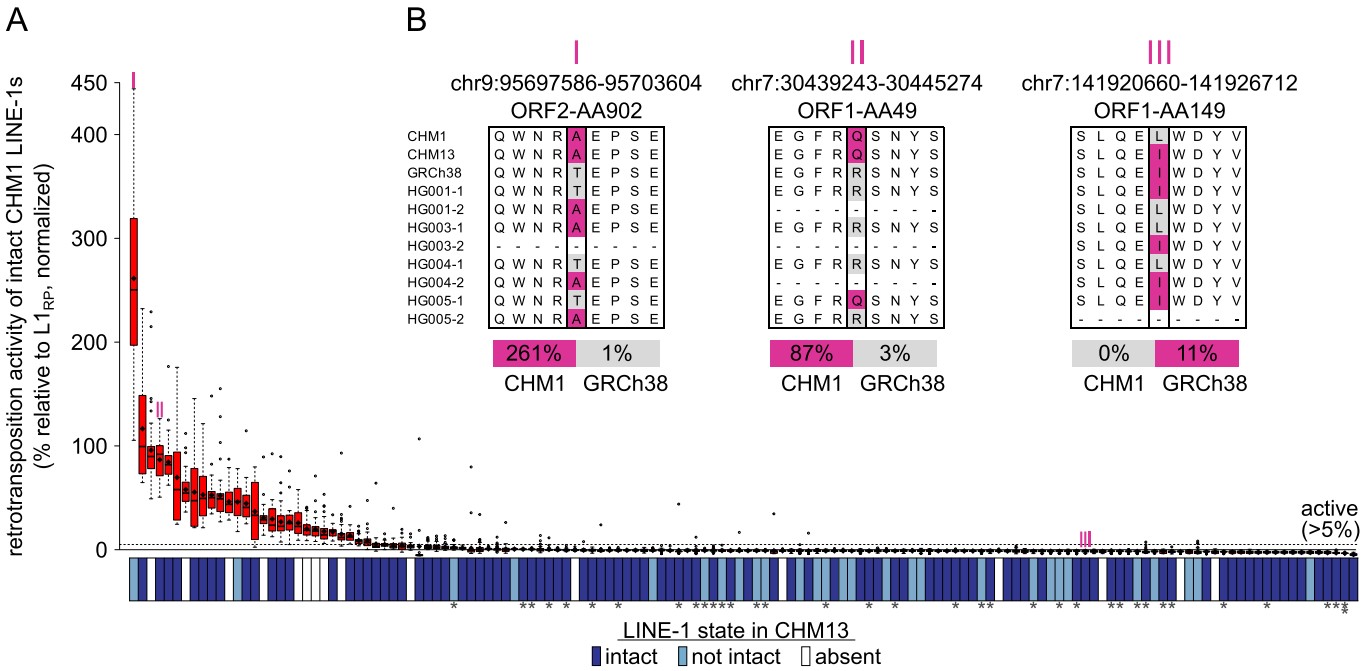

**Figure 2. In vitro retrotransposition activity of all intact LINE-1s in a haploid genome.**

(A) Red boxplots show the distribution of retrotransposition activity of all replicates of each intact LINE-1 from the CHM1 genome normalized to the positive (L1RP: 100%) and negative control (empty vector: 0%). Mean activity of each LINE-1 is represented by a black diamond. A LINE-1 is considered "active" in vitro only when its activity is greater than 5% (horizontal dotted line) that of the positive control. Blue/white boxes below each bar shows whether each intact LINE-1 from CHM1 is present/absent and intact/not-intact at the syntenic allele in CHM13. Stars under the blue/white boxes indicate the lineages of LINE-1s: no star for L1HS, one star for L1PA2 and two stars for L1PA3. (B) Examples of candidate amino acid changes responsible for in vitro activity discrepancy between different allelic forms at one example site. "I," "II," and "III" correspond to the marked LINE-1s on (A). Coordinates indicate the location of the corresponding LINE-1 in the GRCh38 reference genome and amino acid locations relative to L1RP. Magenta color indicates that the associated LINE-1 is tested or inferred to be active in vitro. "HG00X" indicates the sample number from the GIAB project, "−1" and "−2" indicate the two alleles from the same diploid genome. Continuous "−"s indicates that LINE-1 is absent from the allele. Source data are available online for this figure.

We hypothesized that the observed differences in activity could arise from activity-modulating sequence variation in these allelic LINE-1s. Indeed, a previous study found extensive variation in the in vitro activity of the alleles of a single LINE-1 from several individuals, suggesting a model in which mutation accumulation gradually increases the probability of inactivating each LINE-1 allele in a population (Seleme et al, 2006). Consistent with this expectation, of the 29 active LINE-1s from CHM1, only 18 were present and intact in CHM13, two were present but not intact in CHM13 and nine were specific to the CHM1 genome and absent from CHM13 (Fig. 2A). To investigate whether such allelic variation in LINE-1s may modulate in vitro retrotransposition activity, we wanted to identify candidate mutations that are shared by alleles with similar in vitro activity profiles, and hence might underlie the in vitro activity discrepancy.

We collected all available allelic forms of several known "hot" LINE-1s from CHM1, CHM13, the NCBI nt database (including Brouha et al (2003) and Beck et al (2010)), and the GIAB project (Zook et al, 2016) and compared each pair of alleles for which our retrotransposition measurement differed from those generated by Brouha et al and/or Beck et al (Dataset EV1, column AB and AC). In total, there were nine LINE-1s with activity discrepancy between our measurements and another study (five cases where we found activity in LINE-1s measured as inactive previously and two cases

where we measured no activity for a LINE-1 previously reported to be active in vitro). We defined surrogate sites of in vitro activity discrepancy as sites with non-synonymous changes found only in the less active allele (Fig. 2B and Dataset EV2), absent from all known active alleles of that locus. Using these pairs, we defined one or two sites in each pair that fulfilled these criteria which we used as surrogates of their in vitro activity (Dataset EV2). We found that the presence of both in vitro active and inactive LINE-1 alleles at individual loci is common in the population (consistent with a slow decay of LINE-1 sequences after their insertion) (Seleme et al, 2006), and the allele with high in vitro activity was typically the minor allele. These "hidden" active LINE-1 alleles show that the activity of some apparently dead LINE-1s likely persists longer than previously appreciated since each person has a different set of active LINE-1 alleles. These data also suggest that determination of the active LINE-1s from one genome is not sufficient to accurately predict the activity of LINE-1s at the same locations in another individual. Rather, each genome likely harbors a number of LINE-1s that exist in their active state in only a small fraction of individuals (the rare/minor allele). In this way, non-synonymous changes in intact allelic insertions contribute to the variation in the active set of LINE-1s even between two closely related individuals. This high variation in the presence of active LINE-1s between closely related genomes further supports the notion that the

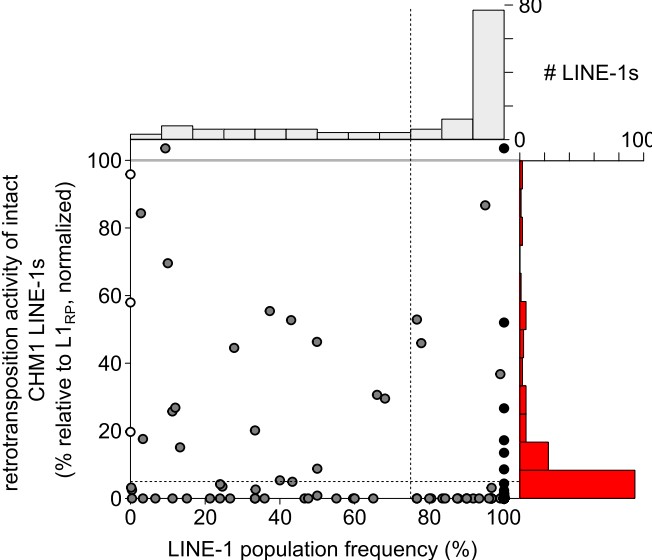

**Figure 3.  Comparison of in vitro activity and population frequency of intact CHM1 LINE-1s.**

The scatter plot shows a dot for each LINE-1, placed according to its in vitro activity and population frequency. LINE-1s that are fixed in the human population are plotted in the "fixed" bin (black dots). White dots represent LINE-1s specific to CHM1, and gray dots represent polymorphic LINE-1s. LINE-1s with higher activity than L1$_{RP}$ were binned together and plotted above the gray line. LINE-1 with activity values not significantly different than a negative control were plotted as zero. Top and right, histograms of population frequencies and retrotransposition activity of CHM1 intact LINE-1s.

repertoire of active LINE-1s varies greatly among human individuals (Streva et al, 2015; Ewing and Kazazian, 2010). It remains unclear how varied the number of active LINE-1s may be across more diverse genomes (Ewing and Kazazian, 2010).

## Frequency spectrum of all LINE-1s in a single genome

The bulk of retrotransposition activity in human genomes is thought to originate from young, polymorphic LINE-1s (Brouha et al, 2003). All 29 of the in vitro active LINE-1s we identified in CHM1 belong to the most recently expanded subgroups of the L1HS (human-specific LINE-1s) family (Fig. 2A). L1HS-Ta1d, the youngest subfamily (Boissinot et al, 2000), accounts for the largest fraction (14/29) of in vitro active elements e also detected in vitro activity from each of the older subfamilies of L1HS: Ta1nd (3/29), Ta0 (7/29), and nonTa (5/29). We also observed in vitro active LINE-1s from older L1HS subfamilies challenging the previous observations that the majority of ongoing LINE-1 retrotransposition potential comes from the youngest subfamilies.

Within the L1HS elements, the active elements are typically assumed to be the youngest, polymorphic LINE-1s. Given our in vitro activity and sequence data for so many active LINE-1s, we could more rigorously look for correlations between the frequency of each LINE-1 in humans and their in vitro activity. Specifically, we determined the "allele frequency" in the human population of the set of intact LINE-1s found in the CHM1 and CHM13 genomes. We integrated data from the 1000 Genomes Project (KGP) (Auton et al, 2015; Abecasis et al, 2012), euL1db (Mir et al,

2015), and two previously published studies (Iskow et al, 2010; Wong et al, 2013) (Dataset EV1, column U). From this survey, we were able to estimate the population frequency of all 142 intact CHM1 LINE-1s we had tested for in vitro activity. More than half of the LINE-1s that are intact in CHM1 are fixed in the human population (74/142), and seven are found only in CHM1, not in CHM13, suggesting they are extremely young; the remaining active LINE-1s are relatively evenly split amongst the intermediate population frequency bins (Fig. 3, top histogram).

Next, we compared these frequency estimates to our in vitro activity data. We found that more than half (77/142) of all intact LINE-1s in CHM1 are both fixed in humans (Dataset EV1, column U, 100% frequency) and inactive in vitro (Fig. 3; rightmost bar of top histogram). Overall, we find LINE-1s in every frequency bin that retain in vitro retrotransposition activity, and the most highly active LINE-1s are either very low frequency or fixed in humans. In fact, we found six fixed LINE-1s that were active in vitro (Fig. 3). Thus, while many of the retrotransposition-competent LINE-1s in each human genome are young and polymorphic (consistent with the prevailing model), a substantial number of old and high frequency LINE-1s also retain significant in vitro activity.

### Measuring the history of LINE-1 in vivo activity

6pt?>Evolutionary successful LINE-1s must retain in vitro activity, but activity in vivo and the ability to produce new copies in the germline genome (hereafter referred to as in vivo fitness) imposes additional constraints. We reasoned that LINE-1s with many close relatives were either recently active in vivo or are closely related to a LINE-1 with recent in vivo activity. Therefore, the distribution of the pairwise sequence differences between a LINE-1 and each other full-length LINE-1 in the genome should reflect that LINE-1's and/or its family's recent in vivo activity. Here, we use this measurable distance as a proxy for the in vivo fitness of each intact CHM1 LINE-1. To do this, we measured the Hamming distance from each intact LINE-1 nucleotide sequence in the CHM1 assembly to every other full-length LINE-1 sequence in the genome (including all 9548 sequences greater than 5000 bp, not just intact LINE-1s). The set of pairwise sequence distances for a given LINE-1 exhibited one of three broadly-defined distributions which likely reflect undetectable, ancient, or recent in vivo activity (dark gray, light gray, and orange, respectively; Fig. 4A). The pairwise distances for a given LINE-1 form some combination of three apparent peaks, which represent its close, intermediate, and distant relatives ("young," "mid," "old" bins Fig. 4A). Every LINE-1 is distantly related to every other LINE-1 in the genome, and these relationships are reflected in the peak of pairwise distances greater than 82 nucleotides that appears in every distribution (Fig. 4A, "old" dark gray bin). LINE-1s with a distribution containing only this "old" peak had no close relatives and hence no detectable in vivo activity (Fig. 4A, black histogram). Some LINE-1s had a distribution with a second peak spanning 28-82 nucleotides differences (Fig. 4A, "mid" bin). Older in vivo activity which resulted in detectable but ancient expansion of that LINE-1 or its close relative would generate this pattern (Fig. 4A, light gray histogram). Finally, some LINE-1s had three peaks – an "old" peak, a "mid" peak, and a third peak which comes from close relatives with only 1-27 differences at the nucleotide level (Fig. 4A, "young" bin). The presence of this group of highly similar LINE-1s suggests recent in vivo activity of these LINE-1s or their close relatives in the ancestral lineage of CHM1 (Fig. 4A, orange histogram). We used the count of LINE-1s within this young, close-

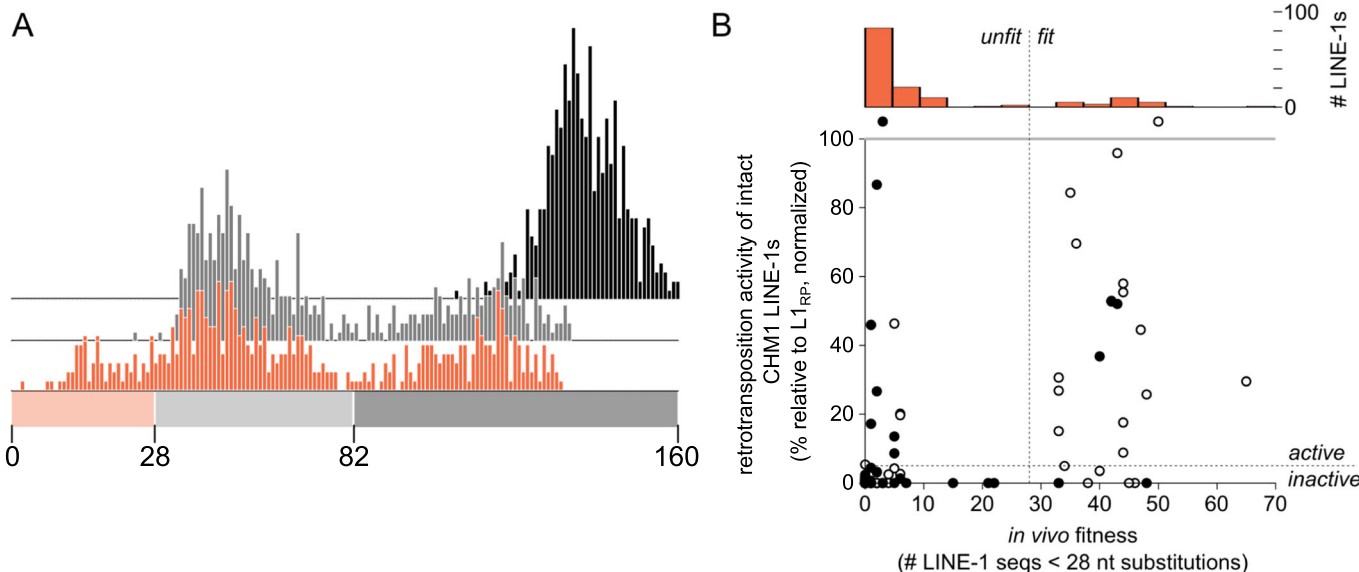

**Figure 4.  A measure of in vivo LINE-1 fitness.**

(A) Exemplar distributions of sequence distance of individual LINE-1s to all other full-length LINE-1s in CHM1, used to infer the in vivo fitness of a LINE-1 and its close relatives. A young (orange), a mid-age (light gray), and an old (dark gray) LINE-1 are shown. (B) Scatter plot of in vitro activity versus in vivo fitness of each intact CHM1 LINE-1. Filled dots represent frequent (>75%) LINE-1s in the human population, and hollow dots represent polymorphic (≤75%) LINE-1s in the human population. LINE-1s with in vitro activity less than or equal to negative control are plotted as zero activity, and both LINE-1s with higher than L1$_{RP}$ activity are shown above the gray line as one category. Dashed lines represent the cutoffs used to call in vitro active/inactive and in vivo fit/unfit. Top, histogram of in vivo fitness of intact LINE-1s in CHM1.

relative-containing region of the distribution as a proxy for the recent in vivo activity of each LINE-1 (Fig. EV4). After calculating this parameter for all intact LINE-1s in the CHM1 genome, we found that most intact LINE-1s have "no near neighbors" (120/154 with "old/black" or "mid/gray" distributions), which suggests that the majority of the intact LINE-1s in this genome have not been active in vivo. Yet, 34/154 LINE-1s show evidence of recent in vivo activity by them or a close relative. These in vivo fit LINE-1s include six LINE-1s we were unable to clone, including 5/7 LINE-1s that are contained within centromeric repeats and cannot be mapped to a BAC.

The capacity for in vitro activity should be prerequisite for in vivo activity (though rare, trans mobilization of inactive elements has been observed (Garcia-Perez et al, 2007; Wei et al, 2001). Consistent with this expectation, the majority of LINE-1s have low in vitro activity and in vivo fitness (bottom left quadrant, Fig. 4B) or high in vitro activity and in vivo fitness (top right quadrant, Fig. 4B). However, we also observed numerous LINE-1s that defy this expectation, suggesting more complex evolutionary histories (detailed in "Discussion"). For example, LINE-1s with high in vivo fitness did not always also have measurable in vitro activity (bottom right quadrant, Fig. 4B). These could be LINE-1s with an inactive allele in CHM1, but which existed in an active and fit allelic state in recent ancestors of CHM1. In contrast, some LINE-1s had measurable (and in some cases very high) in vitro activity (top left quadrant, Fig. 4B), but have not made germline copies in the CHM1 lineage as evidenced by their low in vivo fitness. These LINE-1s are of particular interest, as they could be extremely young and likely to increase their fitness in the near future, or they could be effectively blocked by restriction mechanisms of the extant human genome (see Discussion).

### An analysis of LINE-1 evolutionary history by integration of population frequency, in vitro activity, and in vivo fitness

In addition to the pairwise comparisons of population frequency, in vitro activity, and in vivo fitness presented above, this dataset also allowed us to identify patterns in a 3-way comparison of these parameters that suggested explanations of some outliers within each pairwise comparison. Based on the distribution of each parameter across intact CHM1 LINE-1s, we binarized population frequency into polymorphic/frequent, in vitro activity into active/inactive, and in vivo fitness into fit/unfit. Demonstrating a variety of evolutionary trajectories within the LINE-1s of the CHM1 lineage, intact LINE-1s were spread across all eight possible combinations of these parameters (Fig. 5A). Expectedly, more than half of all intact CHM1 LINE-1s that we successfully cloned were old and dead (84/142). These LINE-1s were fixed or of high frequency, with no detectable in vitro activity, and no closely related LINE-1s in the genome (in vivo unfit). Our pairwise comparison showed that in vitro activity cannot be consistently predicted from in vivo fitness or frequency alone. However, we found that polymorphic in vivo fit LINE-1s are more likely to be active in vitro (15/(15 + 5) polymorphic, fit are in vitro active vs 3/(3 + 5) frequent, fit). These data suggested that population frequency and in vivo fitness (computable parameters) can be used to more accurately predict in vitro active LINE-1s without cell-based retrotransposition data.

In exception to this overall correlation, we observed several outlier LINE-1s with high in vivo fitness but no detectable in vitro activity. Indeed, in vivo fitness derives from the replication history of the collection of LINE-1 alleles at that locus in many genomes, while the retrotransposition assay records the in vitro activity of just one of the alleles at the locus.

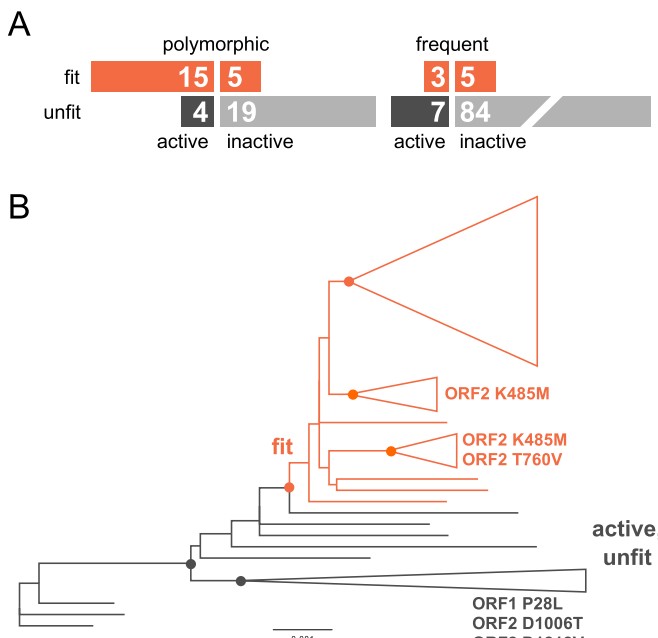

## A

polymorphic

fit | 15 | 5
unfit | 4 | 19
active | inactive

frequent

fit | 3 | 5
unfit | 7 | 84
active | inactive

## B

ORF2 K485M

ORF2 K485M
ORF2 T760V

**fit**

**active, unfit**

ORF1 P28L
ORF2 D1006T
ORF2 D1218V

0.001

**Figure 5. Discrepancy between in vivo fitness and in vitro activity suggests recent adaptations of LINE-1s.**

(A) LINE-1s were categorized according to three binary categories: polymorphic (≤75 % frequency in the human population) or frequent (>75% frequency in the human population); active (>5% in vitro activity of L1$_{RP}$) or inactive (≤5% in vitro activity of L1$_{RP}$); fit in vivo (>10 near neighbors, ≤28 substitutions) or unfit in vivo (≤10 near neighbors, ≤28 substitutions). (B) A maximum likelihood phylogeny of the in vivo unfit LINE-1s and the "in vivo fit and in vitro active" LINE-1s. Dots on the backbone mark the supported nodes (aLRT ≥0.85). Orange dots indicate the supported clusters within the in vivo fit group, and the most basal orange dots indicates the node that separates the two major groups. Labeled non-synonymous substitutions and their ORF coordinates that are specific to the cluster.

Due to the difficulty of making true in vivo fitness measurements (Boissinot et al, 2000), the field has historically used in vitro retrotransposition activity as the primary indicator of activity in humans. However, as our data demonstrates, in vivo fitness requires persistent germline replication of a LINE-1 or related group of LINE-1s over recent human evolution—a phenotype complicated by epigenetics, chromosomal context, and the host innate immune system, amongst other factors. In further support of the nonbinary relationship between in vitro activity and in vivo fitness, we found 11 in vitro active LINE-1s with no evidence of recent in vivo activity (Fig. 5A, the "active" and "unfit" LINE-1s). To test whether there might be sequence changes associated with in vivo fitness, we generated a phylogeny of all the "fit" and "active-unfit" LINE-1s (Fig. 5A, sequences in Dataset EV3). For this analysis, we excluded the 103 inactive and unfit LINE-1s due to the confounding effects of these mostly old and highly mutated sequences. Within the phylogeny of "fit" and "active-unfit" sequences, a single supported node (Fig. 5B, red dot, aLRT >0.9) separated the 28 "fit" LINE-1s from the 11 "active-unfit" LINE-1s. The four phylogenetically informative sites that separate the two groups include two 5'UTR sites and one site in each ORF, all of which show only synonymous changes within the alignment.

We then investigated specific variations shared within the in vivo fit group, distinct from the amino acids shared by the in vivo unfit group. Hypothetically, these phylogenetically informative sites could represent determinants of susceptibility or resistance to host restriction factors. We found that all 28 fit LINE-1s group together within a monophyletic cluster (orange branches, Fig. 5B). Within this group of in vivo "fit" CHM1 LINE-1s, 24 of the 28 fit LINE-1s fell within one of three distinct supported clusters (Fig. 5B, orange dots with triangles representing collapsed branches, aLRT > 0.85). Within two of these clusters, we were able to identify nonsynonymous changes shared by all LINE-1s in the cluster: both clusters contained K485M in ORF2p and one cluster also contained T760V in ORF2p. Although most in vitro active LINE-1s belong to the youngest subfamily of the human-specific LINE-1s, Ta1d (Brouha et al, 2003; Beck et al, 2010; Boissinot et al, 2000), these clusters suggested further diversification within the Ta1d subfamily with yet unknown consequences for potential emergence of new active LINE-1 subgroups.

## Discussion

Decades of research have bridged the first report of an active LINE-1 in humans to extensive databases of LINE-1 insertions in hundreds of human genomes (Brouha et al, 2003; Ewing and Kazazian, 2011; Beck et al, 2010; Iskow et al, 2010; Mir et al, 2015). Many of these datasets address the population genetics of LINE-1s in humans, but many have also analyzed the sequence and in vitro activity (retrotransposition rate) of select LINE-1 mobile element insertions (MEIs) from various genomes (Brouha et al, 2003; Beck et al, 2010). Because LINE-1s are the only measurably active autonomous transposable elements in the human genome, and they are known to be causal or central to a slew of human diseases, comprehensive analyses of active LINE-1s in genomes hold value for understanding and predicting with precision, the "genomic load" of active LINE-1s in each individual. Previous work towards this goal used the LINE-1 sequences represented in the most recent, but incomplete, reference genome assemblies (Brouha et al, 2003). These assemblies were also biased against young LINE-1 insertions by the averaging effect of collapsing multiple genome sequences into a single haploid reference sequence (Beck et al, 2010). In this case, the non-representative genotype of donors neither faithfully represents the complexity found in the human population, nor fully describes the genome of any one individual. Our work addresses several fundamental questions regarding LINE-1 variation and recent adaptation by utilizing high quality haploid and long-read-based genome assemblies. Each LINE-1 in our collection has been previously identified as a structural variant in the CHM genomes (Chaisson et al, 2019). However, the curation of these sequences using a layered approach of RepeatMasker LINE-1 calls followed by in-depth analysis of sequence characteristics and function is a major step towards understanding the functional consequences of the substantial sequence variation in the LINE-1s of individuals. Our data bridges a notable gap in the current understanding of LINE-1 diversity, activity, and evolution in humans: a clear picture of the frequency, allelic diversity, and activity of all the intact LINE-1s in a homozygous human genome.

Our analysis of the long-read-based assemblies of the homozygous CHM genomes revealed LINE-1 variation at multiple levels:

LINE-1s at the same allele can vary by their ORF intactness, sequence, and in vitro activity. Due to the limitations of using short reads to resolve long-repeat sequences, previous studies of LINE-1 variation mostly focused on their presence/absence within the human population. Lutz et al (2003) showed that common LINE-1 alleles at a single locus exhibited up to 16-fold differences in retrotransposition assays. Later, Seleme et al (2006) found activity difference between the LINE-1 alleles at three loci, suggesting that the cumulative retrotransposition rate differs between individuals, in part, because of LINE-1 allelic variation. These observations of allelic variation in activity were also consistent with subsequent studies showing a highly active source element on chromosome 13 was not intact in HGWD but has active and inactive alleles within the human population (Sanchez-Luque et al, 2019). Interestingly, this phenomenon was also observed in non-human primates, suggesting this may be a general property of LINE-1s in genomes (Billon et al, 2022). Our data expand upon these previous studies to include all intact and retrotransposition-active LINE-1s in a homozygous genome We found that approximately one-third of the intact LINE-1s in one CHM genome are present but not intact in the other (Fig. 1A). Further, amongst the LINE-1s that were intact in both CHM genomes, the alleles almost always differed as a result of SNPs and indels (Fig. 1B).

Our in vitro retrotransposition assays showed that 29 CHM1 LINE-1s have activity greater than 5% of the activity of $L1_{RP}$ (26 were above 10%, and 21 were above 20%), compared to the previous estimates of 14 LINE-1s greater than 5% $L1_{RP}$ activity by Brouha et al (2003) based on a draft assembly of the human genome (HGWD)—they reported six "hot" LINE-1s with above 20% $L1_{RP}$ activity, and 11 with above 10% $L1_{RP}$ activity. Given the CHM1 genome is not complicated by LINE-1 hemizygosity or heterozygosity, a simple additive model estimates that 58 LINE-1s with measurable in vitro activity would reside in a related diploid human genome. This number is comparable to the inference of 80–100 active LINE-1s per diploid genome made by Brouha et al (2003). Similar studies of diverse genomes will be required to accurately characterize the true variation in this number among humans.

Although fewer active LINE-1s were found in HGWD, they were not simply a subset of the active LINE-1s found in CHM1: CHM1 and HGWD only share two active LINE-1s. Further evidence of this variation in the set of active LINE-1s among individuals comes from comparing CHM1 and CHM13: 11 of the 29 active CHM1 LINE-1s are either not intact or not present in CHM13 (Fig. 2). Our data showing differing in vitro activity (Fig. 2B and Dataset EV2) among allelic LINE-1s likely explains some of this variation. This also implies that intact and in vitro active allelic forms of LINE-1s may exist for older insertions— several "hot" LINE-1s in CHM1 belong to the non-Ta subfamily, the oldest of the human-specific LINE-1s. Recent structural variation analyses based on haploid human genome assemblies support this finding that a small set of pre-Ta representatives possibly remain active in the human genome (Ebert et al, 2021). Based on the different in vitro activity of LINE-1s alleles at three loci, Seleme et al (2006) proposed a model in which LINE-1s accumulate inactivating mutations over time. Our genome-wide data add further support to this model, with the added subtlety that some active LINE-1s are "hidden" behind the seemingly inactivated allelic forms. This suggests lineages remain active in the human population for much longer than appreciated, with the persistence of active, low-frequency alleles of old LINE-1s.

Previously, the in vitro retrotransposition activity of a LINE-1 has been widely used as a surrogate of retrotransposition rate in the genome. However, differences between in vitro activity and in vivo fitness provide insight to the evolution of LINE-1s in human. For example, epigenetic regulation, variable expression of host restriction factors, and human population stratification could all underlie difference between in vitro activity and in vivo fitness. Previous approaches have used clustering and consensus sequences to estimate in vivo fitness based on repeat copy number (Smit et al, 2013; Gu et al, 2008; Yang et al, 2019b). However, these studies mostly focused on long timescale changes in transposable element sequence and abundance using purely computational methods. Here, we used the distribution of Hamming distances between a given and all other LINE-1s to represent their in vivo fitness (Fig. 4B). This approach is applicable to any genome but requires that the youngest LINE-1s are included in the assembly. Another widely applied approach to approximate the in vivo fitness of LINE-1s is counting the number of 3' transductions generated by specific LINE-1s (Szak et al, 2002), but is limited by the small proportion of LINE-1 retrotranspositions that carry 3' transductions - approximately 15% (Szak et al, 2002). For the 142 intact CHM1 LINE-1s that we tested for in vitro activity, we compared our Hamming-distance-based in vivo fitness to the somatic 3' transduction counts from several cancer genome panels (Tubio et al, 2014; Rodriguez-Martin et al, 2020; Chuang et al, 2021) (Dataset EV1, columns Y, Z and AA). We observed correlation between the two measurements for several LINE-1s (Fig. EV5), but ~17 of our in vivo fitness LINE-1s (Fig. EV5) had few 3' transductions in the existing cancer genome panels. In addition, one well-studied LINE-1 on chromosome 22 with many 3' transductions in cancer genomes (Fig. EV5 and Dataset EV1, line 60) panels did not show any recent in vivo activity in our analysis. These differences could derive from variation of each LINE-1's ability to generate full-length insertions and 3' transductions and difference between the ability of specific LINE-1s to replicate in cancer and germline cells.

We found that the probability that a LINE-1 was active in vitro was much higher if that LINE-1 was in vivo fit and polymorphic in population (Fig. 5A). Among the outliers of this correlation, our data suggests most in vivo fit but in vitro inactive LINE-1s resulted from assaying the inactive (likely rare) allelic form (Fig. 2B). Our phylogenetic analysis suggests an explanation for the other group of outliers – in vivo fit LINE-1s group within a single monophyletic cluster (Fig. 5B, orange dots), while the unfit LINE-1s fall outside of this cluster. This grouping suggests that there may be common sequence features among the unfit LINE-1s (some of which are active in vitro) that render them susceptible to host restriction factors. In addition, the grouping of fit LINE-1s could reflect shared mutations that render these LINE-1s evasive to host regulation and gave rise to the in vivo fit LINE-1 lineage. Indeed, multiple supported phylogenetic clusters are found within the in vivo fit group, suggesting the ongoing adaptation of these LINE-1s. Of note, 11/29 of the in vivo fit LINE-1s in CHM1 are either absent or have a non-intact counterpart in CHM13, suggesting that the adapting LINE-1s are sparsely distributed in the human population.

The presence of two intact ORFs is only one of the necessary conditions for the in vivo retrotransposition of LINE-1. Among these factors, 5'UTR is arguably the most notable one that affects

LINE-1 activity (Philippe et al, 2016; Ewing et al, 2020) and can drive the emergence of new LINE-1 lineages including humans (Swergold, 1990) and other mammals (Naas et al, 1998). Indeed, the distribution of the lengths of LINE-1s in CHM1 and CHM13 clearly shows that there are two peaks around 6100 bp that are approximately 130 bp apart (Figs. EV2 and EV3). These peaks correspond to a deletion in the 5'UTR that occurred at the evolutionary transition between the L1PA3 and L1PA4 subfamilies that significantly affects retrotransposition rate, presumably due to the binding of a host restriction factor (ZNF93) to the deleted region (Jacobs et al, 2014). Additionally, our comparison of in vivo fit LINE-1s against the in vivo unfit but in vitro active LINE-1s (Fig. 5B) reveals two nucleotide changes that are specific to the 5'UTR of the in vivo fit LINE-1s, suggesting that mutations in the 5'UTR are associated with the recent retrotransposition of LINE-1s that cannot be characterized by in vitro retrotransposition assays. Although 5'UTRs have been included in our in vitro retrotransposition assay, the additional strong CMV promoter likely marginalized the effect of the native 5'UTR of LINE-1s. Together, in vitro activity in tissue culture assays is necessary but not sufficient to support in vivo fitness of LINE-1s.

Meanwhile, favorable cellular and genomic context is crucial for the in vivo fitness of LINE-1. We found 11 LINE-1s that are active in vitro but unfit in vivo (Fig. 5A, 4 + 7; Fig. 5B, bottom clade). Because of their in vitro activity, the ORF and 5'UTR sequences of these LINE-1s are unlikely the source of their low in vivo fitness. One possible cause of this low in vivo fitness is the epigenomic regulation of the LINE-1s. It has been shown that LINE-1s can be regulated by histone (Ren et al, 2021) and DNA (Philippe and Cristofari, 2023) modifications. Another potential cause of low in vivo fitness is the presence of proteins that restrict LINE-1 retrotransposition. However, our retrotransposition assays are performed in 293 T cells, which lack expression of known restriction factors (Lin et al, 2014), whereas germline cells express a multitude of restriction factors (Schulz and Harrison, 2019). It follows that the lack of restriction factors in 293 T cells also underlies some cases of discrepancy between in vivo fitness and in vitro activity.

Our study offers a nearly complete investigation of all LINE-1s in a haploid genome. Such investigation allowed us to capture all young LINE-1s in an "actual" genome, rather than the inference based on the "average" reference genome. Although the total number of LINE-1s may appear similar between closely related genomes, the collection of intact (potentially "active" or "functional") LINE-1s can be dramatically different. Given that LINE-1s, particularly the young and polymorphic ones, are known to be either causal to or associated with a slew of human diseases (Payer and Burns, 2019), such variation in LINE-1s could be a major factor to consider when inferring the "risk score" of individuals for these diseases.

# Methods

## Retrieving LINE-1 from the haploid and reference genomes

LINE-1s were identified according to the RepeatMasker annotation from CHM1 and CHM13 assemblies (GenBank assembly accession:

GCA_001297185.1 and GCA_000983455.2). BLAST searches of CHM1 and CHM13 used L1.3 (GenBank accession number: L19088) as a query. The RepeatMasker annotation was filtered for keyword "L1" in the "matching repeat" column and converted to bed format. Subsequently, LINE-1 sequences were retrieved from the genome according to the bed file using the "subseq" function of seqtk (https://github.com/lh3/seqtk). GRCh37 and GRCh38 reference genome sequences and RepeatMasker annotations were downloaded from the annotations of GenBank assembly GCA_001297185.1 and GCA_000983455.2 and processed in a similar manner to the CHM genomes to find LINE-1s. LINE-1s on the ALT contigs of the reference genomes were manually inspected for their corresponding chromosomal location if possible and assigned as an alternative LINE-1 allele of that chromosomal location in the reference genome.

## Intact LINE-1 identification

Intact LINE-1s were identified following our previous protocol (Yang et al, 2019a). Full-length LINE-1s (Files EV1 and EV2) were found by filtering the RepeatMasker annotation of the CHM genomes requiring the length of the annotated LINE-1 sequence to be equal to or longer than 5,000 bp. LINE-1 ORFs were found by using EMBOSS (Rice et al, 2000) "getorf" function on full-length LINE-1s with "-find 1" setting to return the translated sequences of the ORFs. The translated ORFs were subsequently searched using BLASTp with the translated ORFs of L1$_{RP}$ (GenBank accession number: AF148856). For each of the full-length CHM1 and CHM13 LINE-1, a custom perl script processed the BLASTp output to find the ORF of the LINE-1 that forms the longest alignment to the ORF1 and ORF2 protein of L1$_{RP}$. LINE-1s with intact ORFs were identified in the distribution of the longest called ORFs of each LINE-1 that align to the reference ORFs, which correspond to ORF1 length of 338 codons and ORF2 length of 1,275 codons. Singletons near these ORF lengths were manually inspected to find additional ORFs that align to the full length of the L1$_{RP}$ reference ORFs.

## Sequencing of LINE-1s with potential sequencing errors

LINE-1s containing frame-shifting mutations were identified by aligning all annotated LINE-1s in CHM genomes to the protein sequences of L1$_{RP}$ ORFs using the setting of "-F15" of LAST (https://gitlab.com/mcfrith/last). Number of frame-shift mutations were counted for each LINE-1 using a perl script, and LINE-1s with one or two frame-shift mutations were kept for further analyses. Status of each of these LINE-1 were then compared to the status of the LINE-1 in the reference genome (GRCh38) and the other CHM genome. Under the assumption that sequencing errors are unlikely to happen at the same site of different genomes, we focused on re-sequencing the LINE-1 that contain frame-shifting mutations that are not shared with the reference genome or the other CHM genome. LINE-1s were PCR amplified from selected CHM1 BACs that contain the target LINE-1 and only a single LINE-1 (BPRC, https://bacpacresources.org). PCR products were purified and region containing the frame-shift mutilations were sequenced at GENEWIZ. Using the same sequencing strategy, we also sequenced the top five CHM1 LINE-1s in terms of allelic difference to CHM13.

## Synteny of LINE-1 in haploid and reference genomes

CHM1 and CHM13 genome sequences were aligned, chromosome by chromosome, to the GRCh37 and GRCh38 reference genomes using lastz (Harris, 2007) under the setting of "--notransition --step=20 --format=lav." The alignments were then processed using lavToPsl, axtChain, and chainMergeSort functions of the UCSC genome browser utilities (http://hgdownload.soe.ucsc.edu/admin/exe/) with default settings. LINE-1 coordinates on CHM1 and CHM13 were subsequently converted to GRCh37 and GRCh38 coordinates with the default setting of the liftOver tool of the UCSC genome browser utilities and the processed chain file mentioned above. For the LINE-1s that could not be directly lifted-over with the default liftOver setting, we took the sequences that are 2000 bp from each end of LINE-1s and lifted them over to the reference genomes. For this step, the lifted-over coordinates of both extended ends were in the same neighborhood of the target reference genome for almost all LINE-1s. We were unable to obtain coordinates for a small subset of LINE-1s flanked by repetitive sequence using this method, and so are unable to assign a genomic position and to pair possible allelic variants; one exceptional LINE-1 flanked by a repeat-rich region had enough unique sequence on both sides to assign as allelic in the CHM1 and CHM13 genomes but still could not be assigned to a genomic coordinate (Dataset EV1, row 183).

## Population frequencies of intact LINE-1s

To retrieve the frequency of intact CHM1 and CHM13 LINE-1s in the general human population. We utilized data from the 1000 Genomes Project (KGP) (Auton et al, 2015; Abecasis et al, 2012), euL1db (Mir et al, 2015), and two complementary studies (Iskow et al, 2010; Wong et al, 2013). CHM1 and CHM13 LINE-1s were lifted to hg19 coordinates using the UCSC liftover tool. Structural variation calls including the KGP phase 1 (Abecasis et al, 2012), KGP phase 3 (Auton et al, 2015) and complementary studies (Wong et al, 2013) were downloaded from http://dgv.tcag.ca/dgv/docs/GRCh37_hg19_supportingvariants_2016-05-15.txt. Where a LINE-1 is present in the reference genome, the deletion calls that overlapped with the intact CHM LINE-1s were found using the "-f 0.9 -r" setting of the "intersect" function of bedtools (Quinlan and Hall, 2010). Where a LINE-1 is absent in the reference genome, the overlapping insertion calls in KGP were found using the default setting of the "intersect" function of bedtools. Frequency of euL1db (Mir et al, 2015) LINE-1s were calculated by the ratio of the number of individuals to the total individuals included in the corresponding study for each "mrip" insertion. Overlapping LINE-1s between CHMs and euL1db were then identified using the default setting of the "closest" function of bedtools. Because the data shows that most of the non-reference LINE-1s overlapped with mrip entries of euL1db, only LINE-1s with 0 distance between CHM and euL1db were considered the same LINE-1. LINE-1 insertion calls of Iskow et al (2010) were also intersected to the intact CHM LINE-1s using bedtools. A perl script was then applied to filter out the singleton calls that correspond to only one individual in the population. Frequency of LINE-1s were then calculated based on the ratio of individuals that carry the corresponding insertion or deletion to the total number of individuals included in the study. For each LINE-1, when frequency data is available from multiple resources, the resource with a larger population was always used in the later analyses; when a new insertion was not identified in any of the abovementioned

population level studies, LINE-1s were inferred to have <0.1% frequency in the population; when a CHM LINE-1 overlaps with a reference LINE-1 but no deletions in the population, it was inferred to be fixed or have 100% frequency in the population.

## Cloning of intact LINE-1

We isolated DNA from the BACs we had identified as containing the intact LINE-1s, then digested it using the restriction enzyme SalI (NEB). After digestion we amplified the LINE-1s using a set of primers (Dataset EV1, column AD and AE) that allowed us to obtain as much as possible of the full LINE-1 element from each BAC and appended linkers to each end of the products to allow for Gibson assembly into the AscI site of pYX-New-MCS which is a modified version of pYX017 with a BamHI-AscI-BstZ17I cloning site between the existing FseI and PciI sites. This vector allows excision of a LINE-1 using AscI, leaving the backbone of the plasmid including a CMV promoter. PCRs were performed to clone each LINE-1 element of interest including their 5' and 3'UTRs using NEB Q5 polymerase with each 50 μL volume reaction containing 10 μL of 5× Q5 reaction buffer, 1 μL of 10 mM dNTPs, 2.5 μL of 10 μM forward and reverse primers, 1 μL of template DNA (~100 ng/μL), 0.25 μL of Q5 polymerase, and 32.5 μL water. The thermo cycler protocol is: (1) 98 °C for 30 s, (2) 98 °C for 10 s, (3) 49 °C for 30 s, (4) 72 °C for 360 s, (5) repeat steps 2–4 29 additional times, (6) 72 °C for 600 s, (7) hold at 4 °C. We gel purified all successful PCR products using the Zymoclean Gel Purification Kit (Zymo Research), then quantified purified PCR products using a Nanodrop and inserted them into AscI-digested pYX-New-MCS using Gibson assembly via the NEB HiFi DNA Assembly kit. This cloning process created a LINE-1 in the pYX reporter backbone driven by both its native 5'UTR promoter and an upstream CMV promoter. In order to identify assembled vectors that contained the intact LINE-1s we had inserted, we first inserted the products of the Gibson assemblies into *E. coli* (NEB), grew colonies overnight on LB Agar with ampicillin, and isolated 10 colonies per LINE-1. After growing the selected colonies overnight, we isolated the vector, then performed PCR using LINE-1 specific primers to confirm presence of the correct insert within the vector. Three clones of each LINE-1 were picked for in vitro activity assays, and one clone was sequence confirmed using Sanger sequencing with primers that bind in the pYX vector and sequence approximately 500 bp into each end of the LINE-1. Clones of the same LINE-1 with significantly different in vitro activity were sequenced using plasmidsaurus standard whole plasmid sequencing service. Clones with non-synonymous changes compared to the corresponding CHM1 LINE-1 sequence were excluded from further in vitro activity analyses.

## In vitro assay for LINE-1 activity

For each LINE-1 we selected a total of 3 vectors that had been PCR confirmed to have the correct insert to use for our activity assay. On day 1 of the assay, we transfected approximately 200 ng vector into approximately 25,000 293 T cells (ATCC CRL-3216) per well in white 96 well tissue culture plates (Genesee Sci) using the TransIT-LT1 transfection reagent (Mirus Bio) and placed the plates in the incubator overnight. The following day, day 2, we spun down the plates to seat the cells, removed the transfection media, and added 250 μL per well of DMEM with 2.5 μg/μL puromycin. We then

returned the plates to the incubator until day 5, when we removed them from the incubator, spun them to seat the cells, and removed the media. After removal of the media, we added 25 µL Dulbecco's PBS and 25 µL of reagent 1 from the Dual-Glo luciferase kit (Promega Inc.) and mixed well to lyse the cells. After approximately 10 min, we measured *Firefly* luciferase activity with a plate reader. Then we added reagent 2 from the Dual-Glo kit, waited another 10 min, and measured *Renilla* activity.

Each 96 well plate contained 4 wells of positive control (L1$_{RP}$ in pYX017) and 4 wells of negative control (pYX-New-MCS, empty) for standardization and to assess the quality of each plate. Additionally, each plate contained 8 wells each of 11 vectors. We randomized which LINE-1s were present on each plate and the order in which they appeared on the plate, but all wells for any given vector containing a LINE-1 were in the same column on a plate. Each vector containing a LINE-1 was assayed at least 3 times, on at least 2 different plates, and spread out over multiple days to minimize the influence of day and plate effects. In total, each vector containing a LINE-1 was assayed in at least 24 wells, and each LINE-1 was assayed in at least 72 wells (3 vectors containing that LINE-1, each assayed in at least 24 wells). The average luciferase activity reading of the positive and negative controls were taken as 100% and 0% LINE-1 activity reference. Luciferase measurements for each LINE-1 were converted to percent activity based on these references.

### Collection of LINE-1 alleles

LINE-1 alleles were collected by mining the long read sequencing data of the GIAB project (Zook et al, 2016). We used the data of the reference individual (HG001, also known as NA12878), the Ashkenazim trio (HG002, HG003, and HG004) and a Chinese individual (HG005). PacBio reads from each library were aligned to the L1$_{RP}$ reference using BLAST. Any read that hit L1$_{RP}$ were further mapped to each LINE-1 allele of interest (LINE-1 sequence at the locus plus flanking 2 kb). Reads spanning genome-LINE-1 junctions were extracted and aligned for each individual and each LINE-1 allele. Because each individual can have up to two LINE-1 alleles at a given locus, reads at each locus were manually sorted into separate alleles based on their shared changes. Consensus sequence of reads belonging to each LINE-1 allele were taken to represent the allele. LINE-1 alleles were validated in the trio data based on the offspring acquiring an allele from each parent.

### Phylogenetic analysis of LINE-1s

The sequences of the 28 in vivo "fit" LINE-1s and 11 in vivo "fit" but in vitro active LINE-1s were aligned using mafft (Katoh and Standley, 2013) (Dataset EV3). Model selection was performed on the alignment using jModelTest (Posada, 2008) under the AICc criterion, and $GTR + I + G$ was selected. Maximum likelihood phylogeny was made under the selected model using PhyML (Guindon et al, 2010) with aLRT node support.

## Data availability

The nucleotide alignment of selected LINE-1 allele ORFs with discrepant in vitro activities can be found in fasta format in Dataset EV2. For this alignment, ORF1 and ORF2 are separated by a 15 bp gap in the alignment. Sequences are grouped by alleles, separated by empty sequences with their header lines indicating the in vitro activity difference of the upcoming sequences. Sequence names are based on the source of the LINE-1 element: sequences named with chromosome coordinates are from GRCh38, sequences with "LJII" names are from CHM1, sequences with "LBZH" names are from CHM13, sequences with "HG" names are from GIAB genomes with "A1" and "A2" indicating different alleles from the same GIAB individual, and sequences with GenBank accession numbers are from the NCBI BAC/fosmid database (source of Brouha et al and Beck et al sequences). To visualize this file, convert it into fasta format, open in an alignment visualizer that supports translation, then translate the sequences. The alignment of LINE-1s used to identify potentially adaptive mutations (Fig. 5) is available in Dataset EV3. The retrotransposition data shown in Figs. 2A and 3 can be found in Dataset EV1, column U and V. The population frequency data shown in Fig. 3 can be found in Dataset EV1, column W. The bin counts of LINE-1 Hamming distances used to calculate the in vivo fitness terms shown in Fig. 4 can be found in Dataset EV1, columns X, Y, and Z. The 3' transduction counts from previous publications can be found in Dataset EV1, columns AA, AB, and AC. Sequences of intact LINE-1s from CHM1 and CHM13 can be found in fasta format in Dataset EV4 (CHM1) and Dataset EV5 (CHM13).

## Peer review information

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

## Acknowledgements

This work was supported by a grant from the National Institutes of Health (R35GM142733 and R00GM112941 to RNM and T32GM007270 to RPDV via the University of Washington). We thank Miriam Rosenberg, John Huddleston, Holly Wichman, Michael Metzger, members of the Metzger lab, and Janet Young for technical advice and critical comments on this manuscript. We thank Pieter J. de Jong and BACPAC Resources for assistance selecting and obtaining CHM1 BACs. We also thank Review Commons and the guidance of three reviewers for their thoughtful and constructive feedback on this manuscript.

## Author contributions

**Lei Yang**: Data curation; Software; Formal analysis; Validation; Investigation; Visualization; Methodology; Writing—original draft; Writing—review and editing. **Genevieve A Metzger**: Resources; Data curation; Formal analysis; Investigation; Visualization; Methodology; Writing—original draft; Project administration. **Ricky Padilla Del Valle**: Resources; Validation; Investigation; Writing—original draft. **Diego Delgadillo Rubalcaba**: Resources; Validation; Investigation. **Richard N McLaughlin Jr**: Conceptualization; Data curation; Formal analysis; Supervision; Funding acquisition; Validation; Investigation; Visualization; Methodology; Writing—original draft; Project administration; Writing—review and editing.

## Disclosure and competing interests statement

The authors declare no competing interests.

# Expanded View Figures

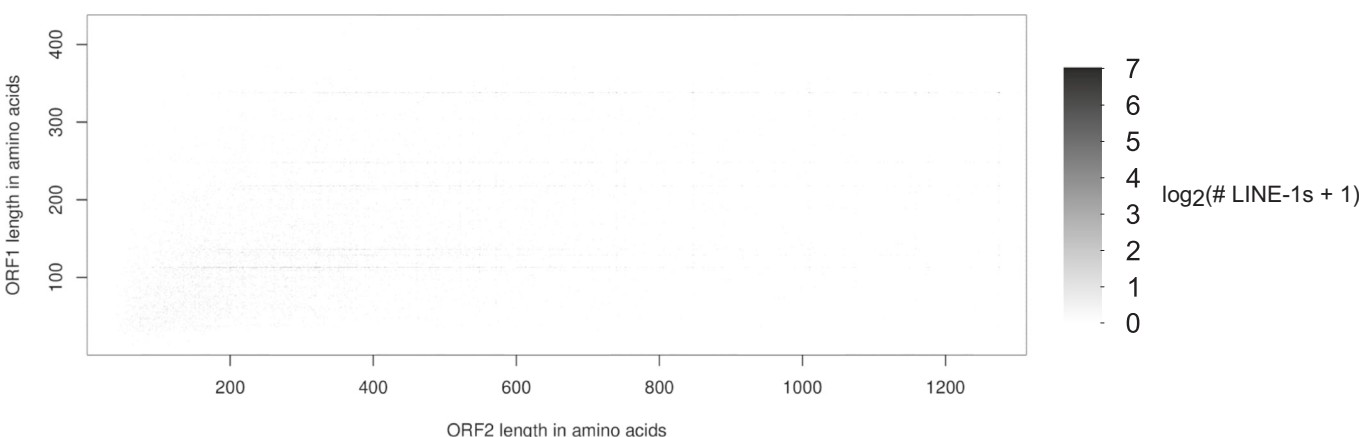

**Figure EV1. Heatmap of translated ORF lengths in full-length LINE-1s from CHM1.**

A zoomed-out version of Fig. 1C shows the distribution of all called translated ORF lengths in the complete set of full-length (>5000 bp) LINE-1 sequences. Many full-length sequences have either an intact ORF1 or ORF2, but only 148 have both intact ORFs that align along the entire length of a reference amino acid sequence (L1$_{RP}$).

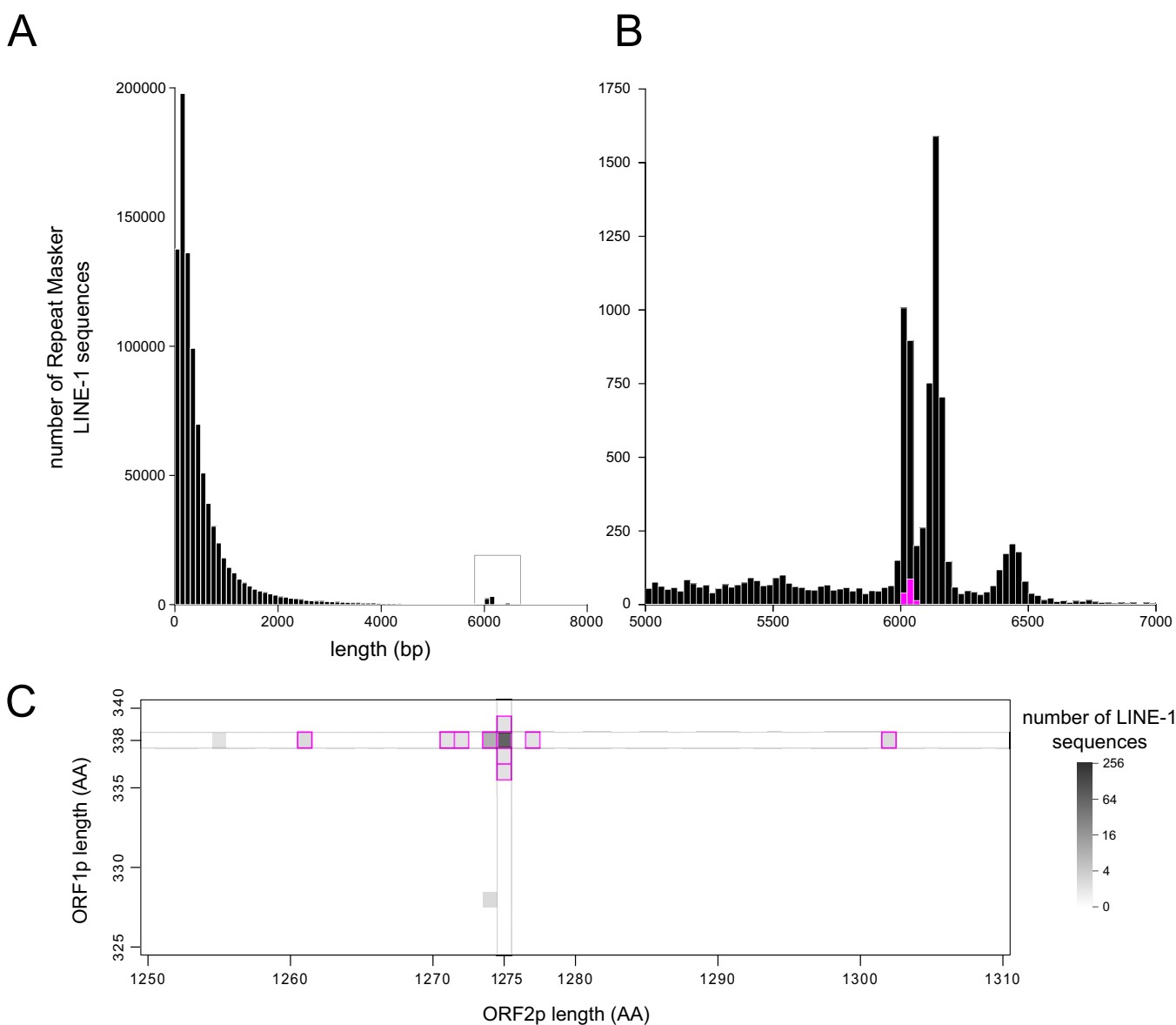

**Figure EV2.  Identification of intact LINE-1s in CHM1.**

(**A**) The distribution of lengths of LINE-1-masked sequences. (**B**) A zoomed-in view of the boxed region of the complete length histogram. Intact LINE-1 sequences are highlighted in magenta. (**C**) A heatmap of the number of sequences with the indicated translated ORF1 and ORF2 lengths. Within the full-length set of LINE-1 sequences, magenta-outlined pixels encode putative ORFs that align along the entire length of the ORF1p and ORF2p sequences of a reference element (L1$_{RP}$).

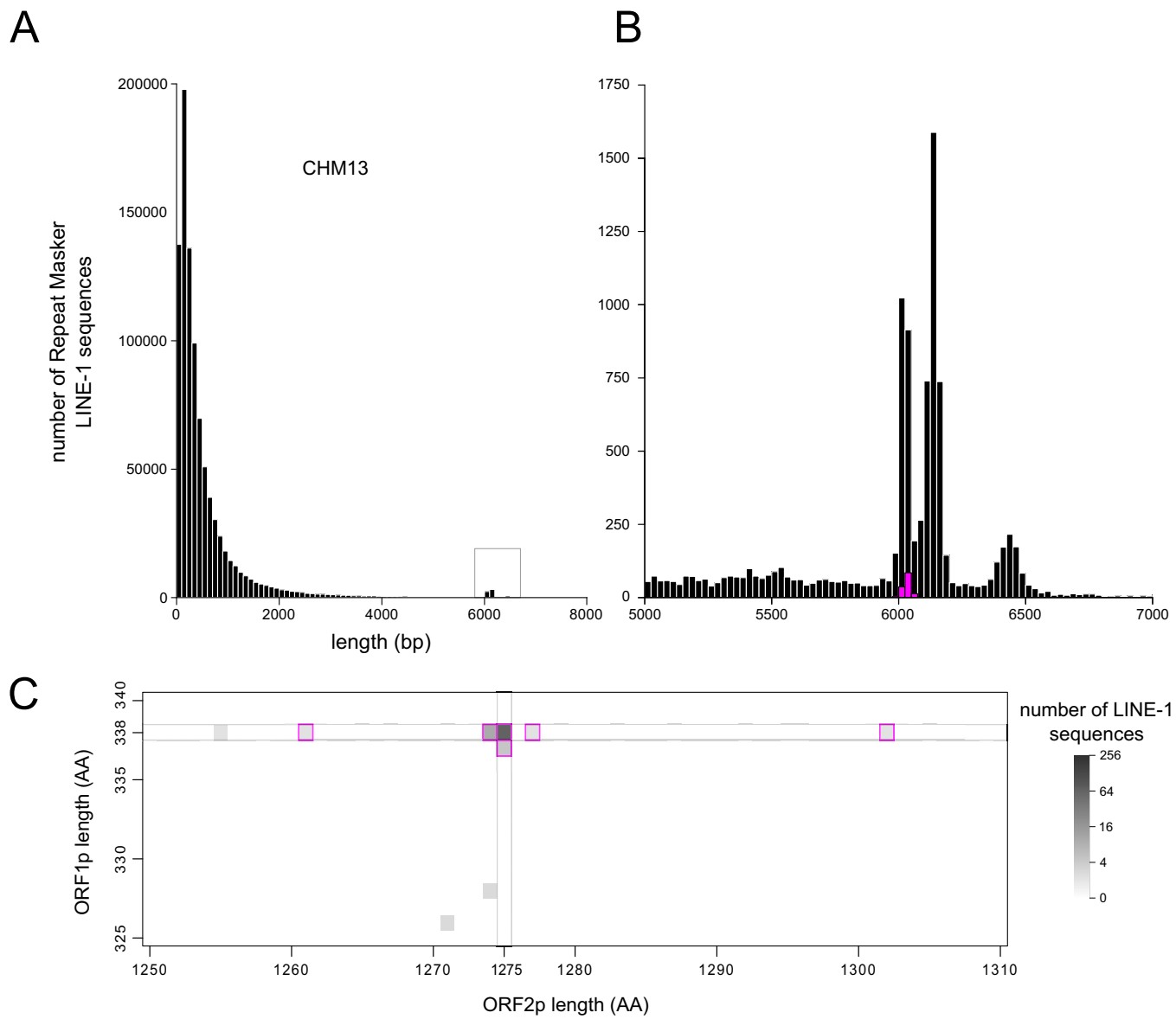

**Figure EV3.   Identification of intact LINE-1s in CHM13.**

(**A**) The distribution of lengths of LINE-1-masked sequences. (**B**) A zoomed-in view of the boxed region of the complete length histogram. Intact LINE-1 sequences are highlighted in magenta. (**C**) A heatmap of the number of sequences with the indicated translated ORF1 and ORF2 lengths. Within the full-length set of LINE-1 sequences, magenta-outlined pixels encode putative ORFs that align along the entire length of the ORF1p and ORF2p sequences of a reference element (L1$_{RP}$).

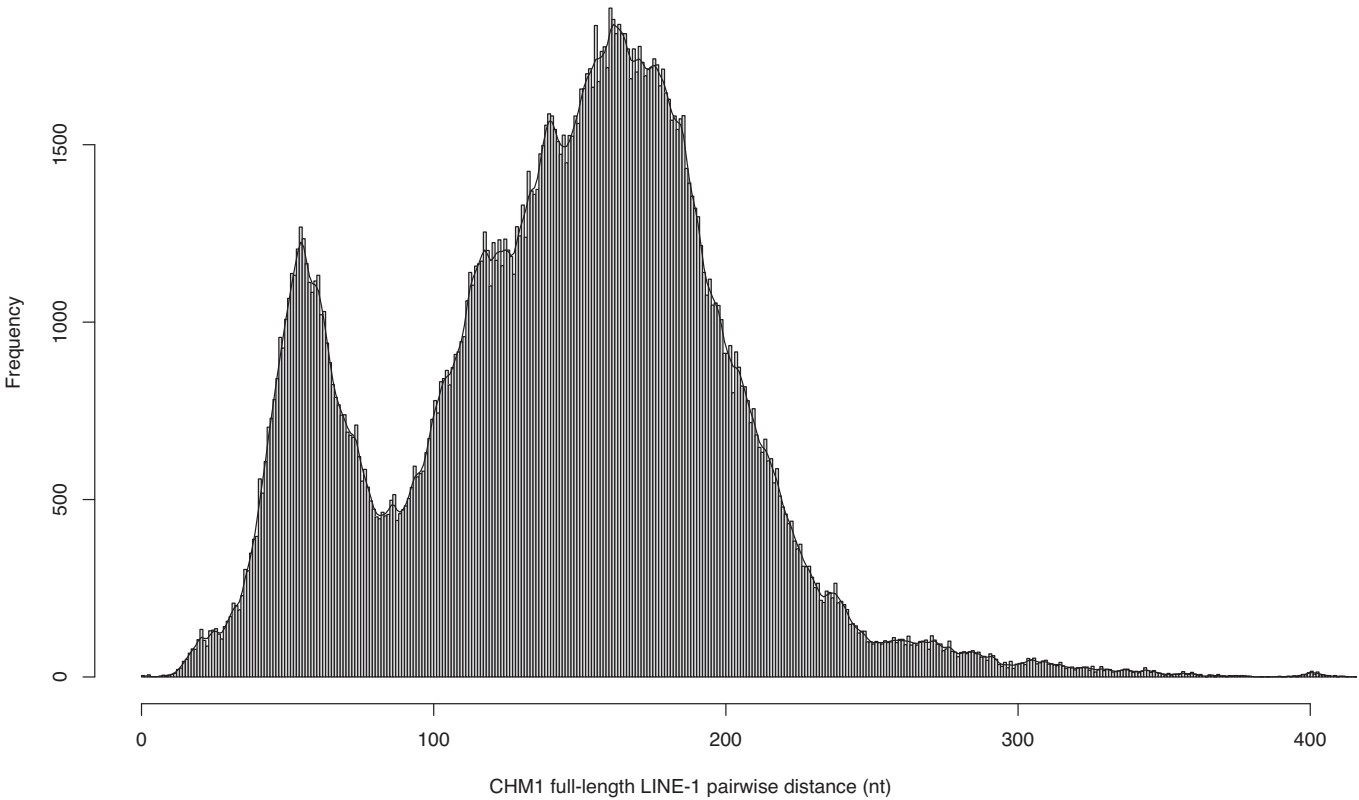

**Figure EV4. Distribution of pairwise distances between full-length LINE-1s in CHM1.**

The histogram shows the frequency of bins of nucleotide distances for pairwise comparisons of all full-length LINE-1 pairs in CHM1.

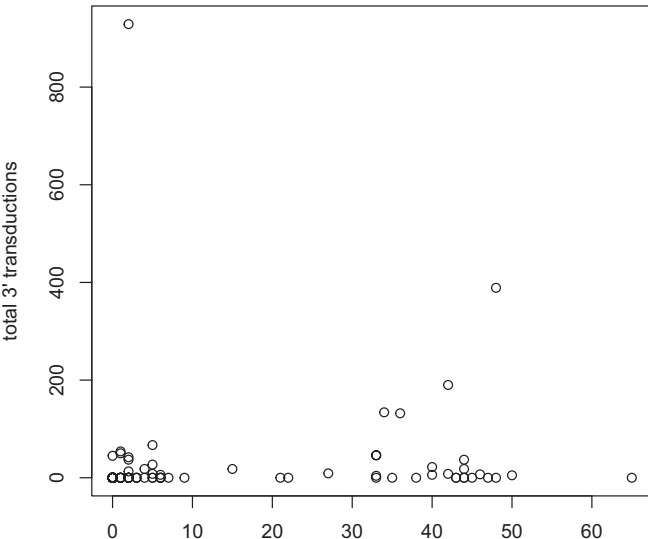

**Figure EV5.   Scatterplot comparison of in vivo fitness and 3′ transductions of intact CHM1 LINE-1s.**

In vivo fitness of a LINE-1 is quantified as the number of full-length LINE-1s within 28 nucleotides distance to that LINE-1. The number of 3′ transductions is the combined number of full-length LINE-1s reported by Tubio et al (2014), Rodriguez-Martin et al (2020), and Chuang et al (2021).

