## [Peer Review File · The EMBO Journal]

Evolutionary insights from profiling LINE-1 activity at allelic resolution in a single human genome

Lei Yang, Genevieve Metzger, Ricky Padilla Del Valle, Diego Delgadillo Rubalcaba, and Richard McLaughlin
DOI: 10.15252/emj.2023115559

Corresponding author: Richard McLaughlin (rmclaughlin@pnri.org)

Review Timeline:

Transfer Date from Review Commons:	8th Sep 23
Editorial Decision:	4th Oct 23
Revision Received:	18th Oct 23
Accepted:	10th Nov 23

Editor: Kelly Anderson

Transaction Report:

This manuscript was transferred to The EMBO Journal following peer review at Review Commons.

Review #1

1. Evidence, reproducibility and clarity:

Evidence, reproducibility and clarity (Required)

Summary:

Yang et al. took advantage of recently published long-read-based genomic sequences of nearly homozygous genomes from complete hydatidiform moles to retrieve allelic sequences of LINE-1, the currently only active and autonomous retrotransposon of the human genome, and produced the repertoire of intact LINE-1 in a genome. The authors performed cell-culture-based retrotransposition assays measurements and in vivo fitness estimations of all identified intact LINE-1 to infer evolutionary dynamics. In this article, the authors further validate the major contribution of polymorphic LINE-1 to the de novo retrotransposition events in the human genome. They also described, at unprecedented resolution, allelic variations among LINE-1 loci and the potential impact of these variations to the interpretation of mutagenic potential of each LINE-1 locus.

Major comments:

1. The key conclusions of the article are mostly convincing. However, it would be a substantial improvement to consolidate the data of the article with information about known active LINE-1s in germ cells or in cancer by using data from recent publications of the Devine and Tubio labs (for example PMID: 34772701, 32024998, 25082706). Across the article, no mention is made of the transductions generated during LINE-1 de novo retrotransposition, which is instrumental to monitor in vivo activity of a group of LINE-1 active copies. It would be of particular interest to make a link between in vitro activity from this study with LINE-1 classification based on their observed activity in cancer (PMID: 32024998, Figure 3b).

2. The use of CHM1 BAC library Sanger sequencing validation and comparison with CHM13 and hg38 sequences is instrumental to support the building of LINE-1 repertoire in CHM1 genome, which is a valuable contribution of the article. The use of a distance-based metric to infer fitness of a LINE-1 is an interesting approach and allow to group LINE-1 copies based on their in vivo activity potential. Again, it would be beneficial to correlate the inferred fitness and retrotransposition activity of copies/alleles, when known, from the above-mentioned literature.

3. Some aspects of the writing of the article should be improved to better support the conclusions.

- In general, the descriptions are dense, and details could be provided in a more direct way to lighten the results section. Several redundancies in the discussion can be combined to increase clarity.

- There is a lack of clarity in the description of how was handled each pair of alleles for which retrotransposition measurements vary between the study and the literature (last paragraph of the "Comprehensive measurement of LINE-1 in vitro activity in a human genome" section). It is not completely clear how the analysis was done and the way the data is presented in File S3 is not

helping to support the conclusion. It could be useful to include some illustrative examples in a panel of Figure 2.

- Regarding inferred in vivo activity, the text contains alternative description with the use of "fit" / "unfit", in vivo "active" / "inactive" or "no closely related LINE-1s" terms. The authors should find a way to clearly define and systematically use one set of terms to enhance clarity along the article. To parallel with in vitro active/inactive, it would be useful to use in vivo fit/unfit.

4. The authors suggest that in vitro activity can be predicted by integration of population frequency and in vivo activity (/fitness) (second paragraph of the "An analysis of LINE-1 evolutionary history [...] and in vivo activity" section). It would be beneficial to strengthen the writing of this section and ultimately validate/test the model by including data from some of the previous studies (e.g. Brouha 2003, Lutz 2003, Seleme 2006, Beck 2010, Rodriguez-Martin 2020, Chuang 2021).

5. The identification of adaptive mutations is only partially described and not strongly supported by experimental or analytical data. It would be interesting to explore the role of phylogenetically informative sites described in Figure 5B/C by testing non CHM1 alleles in retrotransposition assay (by introducing amino acid changes into the cloned CHM1 LINE-1 alleles) or by positioning the sites in ORF1p or ORF2p structure and/or domains to infer impact on functionality.

****Minor comments:****

1. Regarding the in vitro retrotransposition assay, it would be beneficial to provide more data. The current Figure 2 could be enriched by the addition of data related to the variation in the replicates of the experiment (technical but mostly biological with the three clones per LINE-1 tested). Figure 2 could include a dashed line for 100% LIRP and 5% (since it is used as a threshold). It would be useful to provide an additional panel in Figure 2 to illustrate alleles of LINE-1 that are active in this study and compare the values obtained previously in other studies. Similarly, a supplemental table or alignment could be provided to document amino acid changes in the two alleles of each pair (see comment above in the Major Comment 5). The LIHs subfamilies could also be included in the graph of Figure 2 to support the conclusions of remaining active old LIHs at allelic forms in the human genome.

2. Also, the validation of cloning is not well described. The choice of PCR validation must be supported by more technical details on the design of the primers used to validate each copy. The authors should clearly state that the strategy chosen for retrotransposition assay does not rely on the transcription from LINE-1 5UTR but from an upstream strong promoter, ruling out the role of potential mutations in LINE-1 promoter.

3. There are discrepancies with the reported numbers of LINE-1s between Figure 1A and Table S1: 154 vs. 151 in CHM1, 144 vs. 143 in CHM13, respectively.

4. The choice of colors in Figure 3 is not perfectly clear and sometimes not as reported in the text (green highlight and orange highlight). Part of the Figure 3 legend is missing. It should include a description of the color code chosen for the right histogram.

5. For Figure 4, it would be useful to define in the legends the color code for the top histogram. To better read the scatter plot, the words "fit" and "unfit" could be added on each side of the vertical dashed line.

6. In panel B of Figure 5, it seems that the color code and hot/cold description is not fully formatted.

2. Significance:

Significance (Required)

In this article, Yang and colleagues present an unprecedented view of the allelic diversity of young LINE-1 copies related to variable retrotransposition activity in an individual genome. One key aspect of their work is the description of the presence of young active LINE-1 alleles that are absent or non-intact in other genome assemblies, while described at a lower scale in initial work from the Kazazian and Moran labs, cited in the manuscript. The work of Yang et al. demonstrates the requirement of multiple approaches and long-read-based sequencing of individual genomes to fully infer the mutagenesis risk of LINE-1 activity.

The data and methods provided by the authors open the door to a more systematic analysis of mutations and rare allelic forms to understand both mechanistic aspects and evolution of LINE-1 retrotransposition in the human genome. The identification of rare allelic forms of old LINE-1 that retain activity despite previously being considered as inactive is particularly interesting in the light of LINE-1 evolution in the human genome. The authors also describe allelic diversity inside of the Ta1d subfamily, suggesting further diversification and emergence of LINE-1 subgroups. Together with the identification of nucleotide polymorphism among LINE-1 copies, these findings strengthen the notion of individual genomes with individual set of potentially mutagenic LINE-1 alleles.

The findings and methods described in this article are of great interest to a wide audience including the fields of research focusing on human genome evolution, transposable elements, genomic instability, human genetic variation, and personalized medical diagnostic.

Aurélien J. Doucet
CNRS - Université Côte d'Azur

3. How much time do you estimate the authors will need to complete the suggested revisions:

Estimated time to Complete Revisions (Required)

(Decision Recommendation)

Less than 1 month

4. Review Commons values the work of reviewers and encourages them to get credit for their work. Select 'Yes' below to register your reviewing activity at Web of Science

Reviewer Recognition Service (formerly Publons); note that the content of your review will not be visible on Web of Science.

Yes

Review #2

1. Evidence, reproducibility and clarity:

Evidence, reproducibility and clarity (Required)

This manuscript is an interesting and well-crafted study of LINE-1 activity at the single genome human genome level using long read-based haploid assemblies. The manuscript has some real gems and address critical aspects of LINE- biology that are typically not rigorously examined. The authors are to be commended for undertaking this exercise and for providing interesting perspectives that challenge the dogma that dominates the field in several areas. Despite the noted strengths of the contributions, the manuscript ignores the clear limitations inherent to the approaches taken and at times appears as dogmatic as the dogma that they themselves are trying to challenge. These deficiencies should be addressed before this manuscript is published.

Several major and minor points to consider during revision include:

****Major:****

1. Several strategies have been published in the past that have confidently assign LINE-1s to specific loci despite use of shorter reads. These works should be acknowledged, even if as stated in the manuscript, use of longer reads will only continue to add confidence and validity to future assignments.
2. One of the important requirements for precise quantification of LINE-1 activity and predicted risk scores cited in the manuscript was the need to predict activity based on sequence and location. This requirement, as posited in the manuscript, ignores the critical role of epigenetic control in the regulation of LINE-1 activity. As such, a discussion that acknowledges the critical roles of histone and DNA covalent modifications, and that integrates epigenomic insight into predictions of LINE-1 activity must be included in the manuscript.
3. The limitations associated with the use of the CHMI were not addressed in the manuscript. While CHMI contain a paternal only genome, with no maternal contribution, the moles may arise from fertilization of an anuclear empty ovum by a haploid 23,X sperm or fertilization by two sperm giving rise to 46,XX or 46,XY karyotype. As such, generalizable conclusions about CHMI genetics should be carefully made given that the loss of maternal epigenetic imprinting

and gain of paternally imprinted expression may result in abnormal gene expression, including that of LINE-1s. These variances will in turn impact LINE-1 activity profiles.

****Minor****

1. Important citations of previously published work are not properly referenced throughout the manuscript. These are too numerous to identify individually, but the authors should carefully read the manuscript to ensure that proper documentation and reference to previous work is duly acknowledged.
2. There are several typos and missing prepositions that should be corrected. For instance, on page 7, the word "great" should be "greater".

2. Significance:

Significance (Required)

The contribution is highly significant as it challenges previously held concepts and advances our understanding of critical structure and function relationships of Line-1s.

3. How much time do you estimate the authors will need to complete the suggested revisions:

Estimated time to Complete Revisions (Required)

(Decision Recommendation)

Cannot tell / Not applicable

No

Review #3

1. Evidence, reproducibility and clarity:

Evidence, reproducibility and clarity (Required)

Yang et al. perform an in-depth analysis of potentially mobile source L1 alleles in a single human genome (CHM1) previously subjected to Pacbio whole genome sequencing. The retrotransposition efficiencies of source L1 alleles with intact ORFs were tested in vitro, and these efficiencies compared to a model of in vivo activity based on Hamming distance to other ORF-intact L1 alleles. Comparisons of CHM1 L1 alleles are made to CHM13 (used for the recent T2T reference assembly), and also to population-scale sequencing efforts to establish how widespread each source L1 allele is. These data showcase the advantages of being able to resolve L1 alleles with long-read sequencing, allowing the field to make much more accurate predictions of retrotransposition potential in a given genome. The core analyses appear robust and for the most part enough detail is provided to follow what was done.

****Comments:****

1. The text overlooks the potential importance of L1 5'UTR mutations in L1 activity and evolution, as per PMID:25274305, PMID:1701022, and other studies, as well as the impact of genomic context on source L1 activity, as per PMID:27016617, PMID: 33186547 etc. L1 promoter evolution is arguably a major driver of L1 lineage emergence.
2. The way the retrotransposition assay is done here (I think) removes parts of the UTRs as part of introducing L1s into retrotransposition vectors, meaning that the assay tests the biochemical activity of the ORFs. It would be helpful to readers to have a more detailed method for this assay, including the origins of the reporter plasmids, whether there is a CMVp boosting the L1 promoter etc, and some clarity about how much of each L1 was cloned into the assay.
3. Pacbio long-read sequencing has been used previously to locate and characterise L1 alleles in human DNA. The Introduction states: "These represent the first scalable methods to catalog LINE-1 locations and sequences in individual human genomes". The "first" here is questionable. Citations to PMID:31853540 and PMID:34772701 should be included. The latter is particularly relevant as it not only resolves source L1 sequences with PacBio sequencing but also summarises their retrotransposition efficiencies in vitro and population frequencies.
4. I am very interested in the two source L1s (on chr7 and chr9) that were found here to be more active in vitro than L1RP (to my knowledge the most active such element isolated to date, or close to it). Is there anything unusual about these two L1s? A quick look at the supplemental suggested the chr9 element was 5' truncated, was it tested as such in vitro? Also I think it would be worth contrasting the assay (all in HEKs) used here to test efficiency with the assay used by Brouha ... I feel readers may be surprised to find two L1s more mobile than L1RP in one genome.
5. In several places it is mentioned how L1 alleles may differ from sequences provided in reference assemblies, and may therefore explain discrepancies between assay results here and in other studies (e.g. Brouha). The Seleme and Lutz papers are correctly mentioned here, but arguably the most complete demonstration of this concept, from PMID:31230816, is overlooked.

This study reports a chr13 source L1 that was previously found to be inactive by Brouha, and with broken ORFs in the reference genome, has both mobile and immobile alleles in the human population. This L1 is actually in CHM13, but not CHM1, and is "hot" in some individuals and not others. There are several places in the manuscript where this earlier study is very relevant and it would be fair to ask it to be mentioned, especially as the results are concordant. The same concept is reinforced by an even more recent paper (PMID:35728967), except in macaque, showing that this is a general consideration for primate L1 lineages, and actually that source L1 is relatively old and yet jumps extremely well in vitro, which fits an observation made in the present study. Mutually supporting observations like these really add confidence that what is reported in the present study is robust.

6. Hamming distance between ORF-intact source L1 alleles is used to assess in vivo activity. This seems reasonable. However, in other works, transductions have been used to identify families of very closely related L1s. I realise that many highly mobile source L1s will rarely generate insertions carrying transductions, and yet I wonder if any of the youngest L1s in the present study form transduction families, and whether estimates of in vivo activity based on transductions found in population-scale data would reconcile better with in vitro retrotransposition assay data.

7. In the Introduction, it is stated that L1 only transmits vertically. It may be prudent to mildly qualify this position, based on PMID:29983116.

8. A column in Table S2 looks mislabelled: Column R should be CHM1 not CHM13?

Geoff Faulkner (University of Queensland)

2. Significance:

Significance (Required)

This is a well-executed study of considerable interest to the mobile DNA field, and anyone working with long-read DNA sequencing. Its strengths are the genomic and bioinformatic analysis, leveraging the PacBio long-read data and BAC library available for CHM1 to full effect. One limitation (in current form) is its near-exclusive focus on ORFs to encapsulate how mobile a given L1 allele is, when genomic context and L1 promoter mutations could also contribute heavily. Although I liked the manuscript very much and enjoyed reviewing it, some of the conceptual advances are encroached upon by other work (including some very relevant and yet uncited literature). These issues can very likely be addressed via a revision, additional analyses may be required but not new experiments.

Geoff Faulkner (University of Queensland)

3. How much time do you estimate the authors will need to complete the suggested revisions:

Estimated time to Complete Revisions (Required)

(Decision Recommendation)

Between 1 and 3 months

No

We would like to thank the Review Commons editor and three reviewers for their enthusiastic response, including their constructive suggestions and appreciation of the high impact and originality of our study. We have completed the revisions and new analyses suggested by the reviewers, and we thank the reviewers for their suggestions to increase the impact and interest in this work and for guiding us towards this much improved manuscript.

In this response letter, we present the revisions made to the text and figures (purple text) in response to each point from the reviewers (black text).

Reviewer #1 (Evidence, reproducibility and clarity (Required)):

Summary:

Yang et al. took advantage of recently published long-read-based genomic sequences of nearly homozygous genomes from complete hydatidiform moles to retrieve allelic sequences of LINE-1, the currently only active and autonomous retrotransposon of the human genome, and produced the repertoire of intact LINE-1 in a genome. The authors performed cell-culture-based retrotransposition assays measurements and in vivo fitness estimations of all identified intact LINE-1 to infer evolutionary dynamics. In this article, the authors further validate the major contribution of polymorphic LINE-1 to the de novo retrotransposition events in the human genome. They also described, at unprecedented resolution, allelic variations among LINE-1 loci and the potential impact of these variations to the interpretation of mutagenic potential of each LINE-1 locus.

Major comments:

1 - The key conclusions of the article are mostly convincing. However, it would be a substantial improvement to consolidate the data of the article with information about known active LINE-1s in germ cells or in cancer by using data from recent publications of the Devine and Tubio labs (for example PMID: 34772701, 32024998, 25082706). Across the article, no mention is made of the transductions generated during LINE-1 de novo retrotransposition, which is instrumental to monitor in vivo activity of a group of LINE-1 active copies. It would be of particular interest to make a link between in vitro activity from this study with LINE-1 classification based on their observed activity in cancer (PMID: 32024998, Figure 3b).

We thank this and the other reviewers for this suggestion. We agree that a more explicit comparison to the often-reported counts of 3' transductions would be a valuable addition to our analyses. We have added the 3' transduction counts from PMID:34772701, PMID:32024998 and PMID:25082706 to Table S2 (column Y, Z and AA), and made a comparison between these data and our Hamming-distance-based in vivo activity, as the new Figure S5. We found correlations between the two measurements in a significant proportion of LINE-1s, but some interesting exceptions exist which likely reflects the fact that most catalogued 3' transductions come from cancer genomes, and cancer and germline cells represent distinct cellular environments in which distinct sets of LINE-1s are able to replicate (and leave 3' transductions).

In addition to the new figure (Figure S5), we have added a discussion paragraph focused on this interesting comparison.

2 - The use of CHM1 BAC library Sanger sequencing validation and comparison with CHM13 and hg38 sequences is instrumental to support the building of LINE-1 repertoire in CHM1 genome, which is a valuable contribution of the article. The use of a distance-based metric to infer fitness of a LINE-1 is an interesting approach and allow to group LINE-1 copies based on their in vivo activity potential. Again, it would be beneficial to correlate the inferred fitness and retrotransposition activity of copies/alleles, when known, from the above-mentioned literature.

The sequence validation of LINE-1 sequences in CHM1 is an important point which we have addressed in the edited manuscript. Specifically, we used three forms of sequence validation including end-sequencing of one clone of each LINE-1 after it was cloned into the retrotransposition vector and whole-plasmid sequencing of select LINE-1s with discrepant activity amongst the three clones we assayed. In addition, we sequenced the entire LINE-1 sequence for four LINE-1s which had the largest number of mutations relative to their allelic counterpart in CHM13. Please see the above response to 'Major comment 1' for details of our new analysis comparing the previous literature to our data.

3 - Some aspects of the writing of the article should be improved to better support the conclusions.

We thank the reviewer for providing these examples of parts of the text that were particularly difficult to read and comprehend. We have deeply streamlined and improved the text throughout the manuscript based upon detailed editing for readability and clarity by two experienced scientific writers. Below, we detail how we revised the particular sections presented by the reviewer, but we think the entire manuscript is now more succinct and clearer.

- In general, the descriptions are dense, and details could be provided in a more direct way to lighten the results section. Several redundancies in the discussion can be combined to increase clarity.

We have spent considerable time tightening up the text, including removing several overlapping sections from the discussion which can be seen in the included version with changes tracked.

- There is a lack of clarity in the description of how was handled each pair of alleles for which retrotransposition measurements vary between the study and the literature (last paragraph of the "Comprehensive measurement of LINE-1 in vitro activity in a human genome" section). It is not completely clear how the analysis was done and the way the data is presented in File S3 is not helping to support the conclusion. It could be useful to include some illustrative examples in a panel of Figure 2.

We agree that this description was hard to parse, and we have rewritten this and accompanying methods to simplify our explanation of these results. In addition, we have revised Figure 2 to show the data in much more detail. To further aid the logic flow related to this section, we

moved the previous Figure 5B to Figure 2B, updated it with more suitable examples and edited the associated descriptions.

- Regarding inferred *in vivo* activity, the text contains alternative description with the use of "fit" / "unfit", *in vivo* "active" / "inactive" or "no closely related LINE-1s" terms. The authors should find a way to clearly define and systematically use one set of terms to enhance clarity along the article. To parallel with *in vitro* active/inactive, it would be useful to use *in vivo* fit/unfit.

We thank the reviewer for this suggestion and agree with their suggested unified use of '*in vivo* fit/unfit'. To clarify and simplify these terms as much as possible, we added detailed explanations of *in vivo* / *in vitro* activity and systematically defined *in vitro* "active/inactive" (page 5, right column, line 50) and *in vivo* "fit/unfit" (page 8, left column, line 26) at their first appearance in the article, and we changed most instances of "*in vivo* activity" to "*in vivo* fitness" when context permits.

4 - The authors suggest that *in vitro* activity can be predicted by integration of population frequency and *in vivo* activity (/fitness) (second paragraph of the "An analysis of LINE-1 evolutionary history [...] and *in vivo* activity" section). It would be beneficial to strengthen the writing of this section and ultimately validate/test the model by including data from some of the previous studies (e.g. Brouha 2003, Lutz 2003, Seleme 2006, Beck 2010, Rodriguez-Martin 2020, Chuang 2021).

We have thoroughly revised this section of the results (see response to 'Major comment 3' above), per the reviewers suggestion, to increase reader comprehension of this important data. In addition, we greatly appreciate the reviewer's suggestion of a very interesting experimental direction – moving beyond a single long-read-based genome to many diverse genomes, and ultimately calculating the *in vivo* fitness of the LINE-1s from these diverse genomes. For a long time this has not been possible, but the recent publication of the Human Pangenome presents an opportunity to study this interesting question. Though beyond the scope of this paper, our lab is actively working on this fascinating question, and we appreciate the reviewer's shared interest in this question.

5 - The identification of adaptive mutations is only partially described and not strongly supported by experimental or analytical data. It would be interesting to explore the role of phylogenetically informative sites described in Figure 5B/C by testing non CHM1 alleles in retrotransposition assay (by introducing amino acid changes into the cloned CHM1 LINE-1 alleles) or by positioning the sites in ORF1p or ORF2p structure and/or domains to infer impact on functionality.

The reviewer rightly points out that this is one of the most interesting and novel findings of our manuscript. However, the testing of potentially adaptive mutations is potentially complicated and nuanced. Specifically, we don't know the mechanism by which these mutations might be adaptive. It is possible that they simply increase *in vivo* germline retrotransposition activity and this increase would be reflected by an increase of *in vitro* retrotransposition activity. However, another possibility is that these adaptive phenotypes only show themselves *in vivo* or in the context of the host restriction factors expressed in the germline. We strongly agree with the

reviewer that experimental and analytical data on the phylogenetic informative sites associated with the Figure 5 phylogeny is the key to finding out the mechanisms for these changes to affect LINE-1 activity/fitness, and we are, indeed, exploring this very question in the lab now with related projects. We respectfully suggest that these (extremely cool) experiments are beyond scope of this paper, but we have also added some more detailed description and analyses of the potentially adaptive LINE-1 variations from Figure 5 (from page 9, right column, line 50 to page 10, left column, line 5).

Minor comments:

1 - Regarding the *in vitro* retrotransposition assay, it would be beneficial to provide more data. The current Figure 2 could be enriched by the addition of data related to the variation in the replicates of the experiment (technical but mostly biological with the three clones per LINE-1 tested). Figure 2 could include a dashed line for 100% L1RP and 5% (since it is used as a threshold). It would be useful to provide an additional panel in Figure 2 to illustrate alleles of LINE-1 that are active in this study and compare the values obtained previously in other studies. Similarly, a supplemental table or alignment could be provided to document amino acid changes in the two alleles of each pair (see comment above in the Major Comment 5). The L1Hs subfamilies could also be included in the graph of Figure 2 to support the conclusions of remaining active old L1Hs at allelic forms in the human genome.

Upon consideration of this helpful comment, we now augment the presentation of our *in vitro* activity data with a remade Figure 2 with boxplots to show the variation of the data, as well as a horizontal dashed line showing the active-cutoffs and star signs showing which LINE-1s belong to L1Hs or L1PA2.

2 - Also, the validation of cloning is not well described. The choice of PCR validation must be supported by more technical details on the design of the primers used to validate each copy. The authors should clearly state that the strategy chosen for retrotransposition assay does not rely on the transcription from LINE-1 5UTR but from an upstream strong promoter, ruling out the role of potential mutations in LINE-1 promoter.

As detailed above in the response to 'Major Comment 1', we used a combination of end sequencing, whole plasmid sequencing, and multi-read Sanger sequencing to validate the sequences of each LINE-1 cloned from a CHM1 clone. When cloning each LINE-1, we used a specific set of primers designed for the ends of the UTRs for each LINE-1. We have updated the methods and text to clarify this cloning step, and the sequences of these oligos are included in Table S2.

To clarify the fact that our retrotransposition assays use a common, strong promoter, we added text in several places stating this setup and discussing (paragraph that starts at page 11, right column, line 18) how 5'UTRs and other non-ORF factors can affect the rate of LINE-1 *in vitro* activity.

3 - There are discrepancies with the reported numbers of LINE-1s between Figure 1A and Table

S1: 154 vs. 151 in CHM1, 144 vs. 143 in CHM13, respectively.

We thank the reviewer for spotting this error on our part. The numbers in Figure 1 and the main text were correct, and we have revised Table S1 to reflect this data.

4 - The choice of colors in Figure 3 is not perfectly clear and sometimes not as reported in the text (green highlight and orange highlight). Part of the Figure 3 legend is missing. It should include a description of the color code chosen for the right histogram.

We thank the reviewer for bringing this inconsistency to our attention. Based upon feedback from all reviewers, we have simplified the color scheme in Figure 3 and Figure 5 to focus on the core conclusions of these two figures. Specifically, in Figure 3, we have removed the quadrant shading and more clearly presented the cutoffs of 'polymorphic/high frequency' and '*in vitro* active/inactive' as dashed lines in the scatter plot. In Figure 5, we have simplified to two colors – black for *in vivo* unfit and orange to show the *in vivo* fit LINE-1s which is also used in Figure 4 to show the definition of *in vivo* activity. These updated colors are now defined in the figure legends and main text, and we have made references to these colors consistent throughout.

5 - For Figure 4, it would be useful to define in the legends the color code for the top histogram. To better read the scatter plot, the words "fit" and "unfit" could be added on each side of the vertical dashed line.

We thank the reviewer again for suggestions to improve the clarity of our figures. As mentioned above in 'Minor comment 1', we have removed unnecessary colors including the gradient of the histograms in Figure 3 and Figure 4, since the boundaries of each bin are already defined by the axis labels and tics. As suggested, we have also added 'fit' and 'unfit' labels to the dashed cutoff line in Figure 4 to clarify the meaning of this line.

6 - In panel B of Figure 5, it seems that the color code and hot/cold description is not fully formatted.

This formatting error has been corrected.

Reviewer #1 (Significance (Required)):

In this article, Yang and colleagues present an unprecedented view of the allelic diversity of young LINE-1 copies related to variable retrotransposition activity in an individual genome. One key aspect of their work is the description of the presence of young active LINE-1 alleles that are absent or non-intact in other genome assemblies, while described at a lower scale in initial work from the Kazazian and Moran labs, cited in the manuscript. The work of Yang et al. demonstrates the requirement of multiple approaches and long-read-based sequencing of individual genomes to fully infer the mutagenesis risk of LINE-1 activity.

The data and methods provided by the authors open the door to a more systematic analysis of mutations and rare allelic forms to understand both mechanistic aspects and evolution of LINE-1

retrotransposition in the human genome. The identification of rare allelic forms of old LINE-1 that retain activity despite previously being considered as inactive is particularly interesting in the light of LINE-1 evolution in the human genome. The authors also describe allelic diversity inside of the Ta1d subfamily, suggesting further diversification and emergence of LINE-1 subgroups. Together with the identification of nucleotide polymorphism among LINE-1 copies, these findings strengthen the notion of individual genomes with individual set of potentially mutagenic LINE-1 alleles.

The findings and methods described in this article are of great interest to a wide audience including the fields of research focusing on human genome evolution, transposable elements, genomic instability, human genetic variation, and personalized medical diagnostic.

Aurélien J. Doucet
CNRS - Université Côte d'Azur

Reviewer #2 (Evidence, reproducibility and clarity (Required)):

This manuscript is an interesting and well-crafted study of LINE-1 activity at the single genome human genome level using long read-based haploid assemblies. The manuscript has some real gems and address critical aspects of LINE- biology that are typically not rigorously examined. The authors are to be commended for undertaking this exercise and for providing interesting perspectives that challenge the dogma that dominates the field in several areas. Despite the noted strengths of the contributions, the manuscript ignores the clear limitations inherent to the approaches taken and at times appears as dogmatic as the dogma that they themselves are trying to challenge. These deficiencies should be addressed before this manuscript is published.

We thank Reviewer 2 for their enthusiastic appreciation of the value and innovation of our manuscript. We also thank the reviewer for encouraging us to make careful consideration of the missing references relevant to our findings. We have had two researchers with experience in relevant fields edit our text for both readability, clarity, and proper inclusion of relevant references. We have added these throughout and taken careful effort to replace 'dogmatic' statements with clear presentations of the data and thorough referencing of the relevant literature.

Several major and minor points to consider during revision include:

Major:

1. Several strategies have been published in the past that have confidently assign LINE-1s to specific loci despite use of shorter reads. These works should be acknowledged, even if as stated in the manuscript, use of longer reads will only continue to add confidence and validity to future assignments.

We thank the reviewer for this suggestion, and we apologize for the omission of these important publications. As noted above, we have added numerous relevant references (reference 17-27 in the revised text) throughout the text including previous work that used short reads to confidently assign polymorphic/non-reference LINE-1s to specific loci. For example, we now cite the MELT pipeline to detect de novo L1 insertions with short reads (PMID: 28855259), and Iskow et al. 2010, which detects LINE-1s with junction fragment sequencing (PMID: 20603005). We have also added additional text to clarify that short reads are, indeed, often sufficient to place new LINE-1 insertions, while long reads are especially useful for resolving the sequence and location of these insertions. The new text (page 2, left column, line 22-30) presents the advantages/disadvantages of both short reads and long reads.

2. One of the important requirements for precise quantification of LINE-1 activity and predicted risk scores cited in the manuscript was the need to predict activity based on sequence and location. This requirement, as posited in the manuscript, ignores the critical role of epigenetic control in the regulation of LINE-1 activity. As such, a discussion that acknowledges the critical roles of histone and DNA covalent modifications, and that integrates epigenomic insight into predictions of LINE-1 activity must be included in the manuscript.

We thank the reviewer for suggesting this important discussion point. In response, we have expanded our discussion of this topic to place our data in the context of other literature on the effects of epigenomic regulation on in vivo LINE-1 activity, including histone and DNA modifications, as well as the effects of post transcriptional restriction factors (paragraph starting at page 11, right column, line 42).

3. The limitations associated with the use of the CHMI were not addressed in the manuscript. While CHMI contain a paternal only genome, with no maternal contribution, the moles may arise from fertilization of an anuclear empty ovum by a haploid 23,X sperm or fertilization by two sperm giving rise to 46,XX or 46,XY karyotype. As such, generalizable conclusions about CHMI genetics should be carefully made given that the loss of maternal epigenetic imprinting and gain of paternally imprinted expression may result in abnormal gene expression, including that of LINE-1s. These variances will in turn impact LINE-1 activity profiles.

We thank the reviewer for pointing out this confusingly written section of our manuscript, and we agree with the reviewer that LINE-1 activity measurements could be complicated in the CHM cell lines; however, all of our retrotransposition assays were carried out in the common background of 293T cells (chosen because of their low expression of know LINE-1 restriction factors (PMID: 25182477). We have modified the text (page 11, right column, line 52) to clarify these points.

Minor

1. Important citations of previously published work are not properly referenced throughout the manuscript. These are too numerous to identify individually, but the authors should carefully read the manuscript to ensure that proper documentation and reference to previous work is duly acknowledged.

Please see our above response to 'Major point 1'.

2. There are several typos and missing prepositions that should be corrected. For instance, on page 7, the word "great" should be "greater".

Please see our above response to 'Major point 1' and Reviewer 1's 'Major comment 3' for details on our in depth editing of the manuscript.

Reviewer #2 (Significance (Required)):

The contribution is highly significant as it challenges previously held concepts and advances our understanding of critical structure and function relationships of Line-1s.

Reviewer #3 (Evidence, reproducibility and clarity (Required)):

Yang et al. perform an in-depth analysis of potentially mobile source L1 alleles in a single human genome (CHM1) previously subjected to Pacbio whole genome sequencing. The retrotransposition efficiencies of source L1 alleles with intact ORFs were tested *in vitro*, and these efficiencies compared to a model of *in vivo* activity based on Hamming distance to other ORF-intact L1 alleles. Comparisons of CHM1 L1 alleles are made to CHM13 (used for the recent T2T reference assembly), and also to population-scale sequencing efforts to establish how widespread each source L1 allele is. These data showcase the advantages of being able to resolve L1 alleles with long-read sequencing, allowing the field to make much more accurate predictions of retrotransposition potential in a given genome. The core analyses appear robust and for the most part enough detail is provided to follow what was done.

We thank Reviewer 3 for their in depth reading and analysis of our manuscript and data, and for their enthusiasm about the importance of this work in the context of foundational research from their lab and many others in the field. We have carefully considered each comment and completed several new analyses of our data and related data from other publications. We feel that our manuscript is much improved with this new data, as detailed below.

Comments:

1) The text overlooks the potential importance of L1 5'UTR mutations in L1 activity and evolution, as per PMID:25274305, PMID:1701022, and other studies, as well as the impact of genomic context on source L1 activity, as per PMID:27016617, PMID: 33186547 etc. L1 promoter evolution is arguably a major driver of L1 lineage emergence.

We thank the reviewer for suggesting these important additions. To present the relevance of 5'UTR mutations on LINE-1 activity and evolution, we added a discussion paragraph (paragraph starting at page 11, right column, line 16) to address how 5'UTRs and other non-ORF factors can affect the rate of LINE-1 *in vitro* activity. Several key references have been added and discussed in the paragraph: PMID:25274305 reported the regulation of human LINE-1 by the evolution of its 5'UTR; PMID:1701022 was one of the earliest papers that found the effect the

5'UTR promoters on human LINE-1 retrotransposition; PMID: 27016617 and PMID: 33186547 reported specific L1 loci regulated by different promoters and was included in the discussion; PMID:9430649 was one of the examples of non-human LINE-1 lineages emerging because of different promoters and was cited in the added discussion paragraph. We have also added discussion points to make clear that genomic content has a clear role in the activity of source LINE-1s (paragraph starting at page 11, right column, line 42).

2) The way the retrotransposition assay is done here (I think) removes parts of the UTRs as part of introducing L1s into retrotransposition vectors, meaning that the assay tests the biochemical activity of the ORFs. It would be helpful to readers to have a more detailed method for this assay, including the origins of the reporter plasmids, whether there is a CMVp boosting the L1 promoter etc, and some clarity about how much of each L1 was cloned into the assay.

We have added relevant details to the results (page 6, left column, line 5), discussion (page 11, right column, line 52), and methods (page 13, right column, line 16 and 30) sections to clarify the reviewer's important points. The LINE-1s tested for *in vitro* activity were cloned in their entirety (UTRs and ORFs) but driven by both their native promoters in the 5'UTR as well as an upstream CMV promoter. Also, please see our response to Reviewer 1 'Minor comment 2' above.

3) Pacbio long-read sequencing has been used previously to locate and characterise L1 alleles in human DNA. The Introduction states: "These represent the first scalable methods to catalog LINE-1 locations and sequences in individual human genomes". The "first" here is questionable. Citations to PMID:31853540 and PMID:34772701 should be included. The latter is particularly relevant as it not only resolves source L1 sequences with PacBio sequencing but also summarises their retrotransposition efficiencies *in vitro* and population frequencies.

We apologize for leaving out these and other important references, and we agree that the "first" claim is unnecessary. We have added the references suggested for the reviewer as well as several other important references as detailed in the above response to Reviewer 2 'Major point 1'. In addition, we have revised the adjacent text and deleted any references to our work as the "first" in these approaches.

4) I am very interested in the two source L1s (on chr7 and chr9) that were found here to be more active *in vitro* than L1RP (to my knowledge the most active such element isolated to date, or close to it). Is there anything unusual about these two L1s? A quick look at the supplemental suggested the chr9 element was 5' truncated, was it tested as such *in vitro*? Also I think it would be worth contrasting the assay (all in HEKs) used here to test efficiency with the assay used by Brouha ... I feel readers may be surprised to find two L1s more mobile than L1RP in one genome.

To provide more details about the two active L1s (chr7 and chr9), we investigated key changes that could be related to the *in vitro* activity of these elements and now show them in Figure 2B and File S3. In the process of this updated analysis and suggested modifications to Figure 2 by this reviewer and Reviewer 1, we saw that the chr7 L1, mentioned here, had one very high activity measurement pulling its activity above L1RP. As such, we decided to more rigorously

normalize our data by using the positive and negative controls across all plates of each day instead of normalizing to the controls of individual plates, as we had previously done. In addition, for any L1 with discrepant activity among the three clones we assayed, we used whole plasmid sequencing to confirm the identity and consistency of all three clones. In three cases, we found that one or two of the three clones was the wrong L1, and hence excluded them for the *in vitro* activity calculation. After this validation and testing of additional clones, all clones from the same L1 have consistent *in vitro* activity (see updated Figure 2). The updated *in vitro* activity of the chr7 L1 is at 86.7% L1RP, and the chr9 L1 is at 261.4% L1RP in addition to the chr17 LINE-1 with 117% L1RP and two additional LINE-1s that have near-L1RP activity levels (Table S2, column S). These changes in L1 activity were updated in the text, figures, and supplemental materials. Also, we note that the chr9 element is 6019bp in length and was tested as such *in vitro*. Current work in the lab is attempting to understand the mechanisms of increased LINE-1 *in vitro* and *in vivo* activity, as described in detail in response to Reviewer 1's 'Major comment 5'.

5) In several places it is mentioned how L1 alleles may differ from sequences provided in reference assemblies, and may therefore explain discrepancies between assay results here and in other studies (e.g. Brouha). The Seleme and Lutz papers are correctly mentioned here, but arguably the most complete demonstration of this concept, from PMID:31230816, is overlooked. This study reports a chr13 source L1 that was previously found to be inactive by Brouha, and with broken ORFs in the reference genome, has both mobile and immobile alleles in the human population. This L1 is actually in CHM13, but not CHM1, and is "hot" in some individuals and not others. There are several places in the manuscript where this earlier study is very relevant and it would be fair to ask it to be mentioned, especially as the results are concordant. The same concept is reinforced by an even more recent paper (PMID:35728967), except in macaque, showing that this is a general consideration for primate L1 lineages, and actually that source L1 is relatively old and yet jumps extremely well *in vitro*, which fits an observation made in the present study. Mutually supporting observations like these really add confidence that what is reported in the present study is robust.

We thank the reviewer for their suggestion to include these highly relevant and important papers; we apologize for this initial omission. We have now added several sentences to the introduction and discussion (top left paragraph page 11) in addition to citations of these papers.

6) Hamming distance between ORF-intact source L1 alleles is used to assess *in vivo* activity. This seems reasonable. However, in other works, transductions have been used to identify families of very closely related L1s. I realise that many highly mobile source L1s will rarely generate insertions carrying transductions, and yet I wonder if any of the youngest L1s in the present study form transduction families, and whether estimates of *in vivo* activity based on transductions found in population-scale data would reconcile better with *in vitro* retrotransposition assay data.

We thank the reviewer for pointing out our exclusion of data on 3' transductions, the most commonly used surrogates of *in vivo* activity, while also acknowledging that only a small percent of new L1 retrotranspositions carry 3' transduction. Please see our above response to Reviewer 1's 'Major comment 1' for details on our newly added comparison of our *in vivo* activity data to

the 3' transduction-based somatic LINE-1 retrotransposition landscape of those reported in PMID:34772701, PMID:32024998 and PMID:25082706.

7) In the Introduction, it is stated that L1 only transmits vertically. It may be prudent to mildly qualify this position, based on PMID:29983116.

The referenced text in the introduction has been changed from "LINE-1s only transmit vertically" to "LINE-1s generally transmit vertically with few exceptions", with the addition of the suggested citation.

8) A column in Table S2 looks mislabelled: Column R should be CHM1 not CHM13?

We thank the reviewer for seeing this error. Column P (Column R in the previous version) of Table S2 is now correctly labeled as "CHM1 L1 intactness".

Geoff Faulkner (University of Queensland)

Reviewer #3 (Significance (Required)):

This is a well-executed study of considerable interest to the mobile DNA field, and anyone working with long-read DNA sequencing. Its strengths are the genomic and bioinformatic analysis, leveraging the PacBio long-read data and BAC library available for CHM1 to full effect. One limitation (in current form) is its near-exclusive focus on ORFs to encapsulate how mobile a given L1 allele is, when genomic context and L1 promoter mutations could also contribute heavily. Although I liked the manuscript very much and enjoyed reviewing it, some of the conceptual advances are encroached upon by other work (including some very relevant and yet uncited literature). These issues can very likely be addressed via a revision, additional analyses may be required but not new experiments.

Geoff Faulkner (University of Queensland)

Dear Rick,

Congratulations on a great revision! Overall, the referees have been positive. However, there remain a few editorial items that I ask you to address in a revised manuscript as described below:

1. Please provide the EMBO author checklist.
2. Please upload the main figures as individual high resolution figure files, legends must be in the manuscript file, after references. Supplemental figures should also be uploaded as separate figure files, with all panels indicated, renamed "Figure EV1", etc. and their legends should be in the manuscript text after the main figure legends.
3. Up to five keywords, which may or may not appear in the title, should be given in alphabetical order, below the abstract, each separated by a slash (/).
4. Please provide a data availability section as defined in our author guide online.
5. Please enter the author contributions to EJP online.
6. Please review our new policy on conflict of interests on the EMBO author guide online and add this under the title: Disclosure and competing interests statement
7. References should be alphabetical, up to 10 author names can be included followed by et al. DOIs should be removed.
8. Table S1 should be renamed "Table EV1" and Table S2 should be renamed to "Dataset EV1". Legends should be removed from manuscript and added in the excel files.
9. Please include an appendix file with a table of contents.
10. We require source data, please refer to the email from Hannah Sonntag for more information.
11. We include a synopsis of the paper (see <http://emboj.embopress.org/>). Please provide me with a general summary statement and 3-5 bullet points that capture the key findings of the paper.
12. We also need a summary figure for the synopsis. The size should be 550 wide by 200-440 high (pixels). You can also use something from the figures if that is easier.
13. Please provide the files S1-S4 in an appendix in PDF format with a table of contents, and "Appendix Supplementary Sequences S1" etc. Legends should be removed from the manuscript and added to the appendix.

Thank you for the opportunity to consider your work for publication, I look forward to your revision.

Kind regards,

Kelly

Kelly M Anderson, PhD
Editor, The EMBO Journal
k.anderson@embojournal.org

Further information is available in our Guide For Authors: <https://www.embopress.org/page/journal/14602075/>

authorguide

Referee #1:

The authors have addressed all of my comments in full, thank you.

Referee #2:

Summary:

Yang et al. took advantage of recently published long-read-based genomic sequences of nearly homozygous genomes from complete hydatidiform moles to retrieve allelic sequences of LINE-1, the currently only active and autonomous retrotransposon of the human genome, and produced the repertoire of intact LINE-1 in a genome. The authors performed cell-culture-based retrotransposition assays measurements and in vivo fitness estimations of all identified intact LINE-1 to infer evolutionary dynamics. In this article, the authors further validate the major contribution of polymorphic LINE-1 to the de novo retrotransposition events in the human genome. They also described, at unprecedented resolution, allelic variations among LINE-1 loci and the potential impact of these variations to the interpretation of mutagenic potential of each LINE-1 locus.

Comments:

The authors have replied with clarity to my previous requests through the Review Commons process. They provided clarifications in the methods and figures, added references to previous relevant studies and followed requests regarding the inclusion of 3' transductions data.

I really appreciate the quality and thorough improvement of the manuscript along the process. Answers to complementary comments of the two other reviewers are a real benefit for the quality of the work presented by the authors. Though there are still some very minor editing to perform through proofreading of the manuscript, I find it adequate for the EMBO Journal and strongly support its publication.

Rev_Com_number: RC-2022-01774
New_manu_number: EMBOJ-2023-115559
Corr_author: McLaughlin
Title: Evolutionary insights from profiling LINE-1 activity at allelic resolution in a single human genome

The authors addressed the minor editorial issues.

Dear Rick,

Congratulations on an excellent manuscript, I am pleased to inform you that your manuscript has been accepted for publication in The EMBO Journal. Thank you for your comprehensive response to the referee concerns and for providing detailed source data. It has been a pleasure to work with you to get this to the acceptance stage.

I will begin the final checks on your manuscript before submitting to the publisher next week. Once at the publisher, it will take about 3 weeks for your manuscript to be published online. As a reminder, the entire review process, including referee concerns and your point-by-point response, will be available to readers.

I will be in touch throughout the final editorial process until publication. In the meantime, I hope you find time to celebrate!

Warm wishes,
Kelly

Kelly M Anderson, PhD
Editor, The EMBO Journal
k.anderson@embojournal.org
